J Physiol 600.16 (2022) pp 3749–3774

# Oral digoxin effects on exercise performance, K+ regulation and skeletal muscle Na+,K+-ATPase in healthy humans

Simon Sostaric[1] , Aaron C. Petersen[1] , Craig A. Goodman[1,2] , Xiaofei Gong[1], Tai-Juan Aw[3] , Malcolm J. Brown[4] , Andrew Garnham[1] , Collene H. Steward[1] , Kate T. Murphy[1,2] , Kate A. Carey[6] , James Leppik[1] , Steve F. Fraser[5] , David Cameron-Smith[7] , Henry Krum[3,†] , Rodney J. Snow[5] and Michael J. McKenna[1]

[1]Institute for Health and Sport, Victoria University, Melbourne, Australia

[2]Centre for Muscle Research, Department of Anatomy and Physiology, University of Melbourne, Parkville, Australia

[3]Department of Epidemiology and Preventive Medicine, Monash UniversityAlfred Hospital, Melbourne, Australia

[4]Department of Biochemistry and Pharmacology, University of Melbourne, Melbourne, Australia

[5]Institute of Physical Activity and Nutrition, School of Exercise and Nutrition Sciences, Deakin University, Melbourne, Australia

[6]School of Women's and Children's Health, University of New South Wales, Sydney, Australia

[7]College of Engineering, Science and Environment, The University of Newcastle, Ourimbah, NSW, Australia

Handling Editors: Scott Powers & Bruno Grassi

The peer review history is available in the Supporting information section of this article (https://doi.org/10.1113/JP283017#support-information-section).

Simon Sostaric and Aaron C. Petersen are equal first authors.

[†]Deceased.

The Journal of Physiology

**Abstract** We investigated whether digoxin lowered muscle $Na^+,K^+$-ATPase (NKA), impaired muscle performance and exacerbated exercise $K^+$ disturbances. Ten healthy adults ingested digoxin (0.25 mg; DIG) or placebo (CON) for 14 days and performed quadriceps strength and fatiguability, finger flexion (FF, $105\%_{peak-workrate}$, $3 \times 1$ min, fourth bout to fatigue) and leg cycling (LC, 10 min at 33% $V_{O_2peak}$ and 67% $V_{O_2peak}$, 90% $V_{O_2peak}$ to fatigue) trials using a double-blind, crossover, randomised, counter-balanced design. Arterial (a) and antecubital venous (v) blood was sampled (FF, LC) and muscle biopsied (LC, rest, 67% $V_{O_2peak}$, fatigue, 3 h after exercise). In DIG, in resting muscle, [$^3$H]-ouabain binding site content (OB-$F_{ab}$) was unchanged; however, bound-digoxin removal with Digibind revealed total ouabain binding (OB+$F_{ab}$) increased (8.2%, $P = 0.047$), indicating 7.6% NKA–digoxin occupancy. Quadriceps muscle strength declined in DIG ($-4.3\%$, $P = 0.010$) but fatiguability was unchanged. During LC, in DIG (main effects), time to fatigue and $[K^+]_a$ were unchanged, whilst $[K^+]_v$ was lower ($P = 0.042$) and $[K^+]_{a-v}$ greater ($P = 0.004$) than in CON; with exercise (main effects), muscle OB-$F_{ab}$ was increased at 67% $V_{O_2peak}$ (per wet-weight, $P = 0.005$; per protein $P = 0.001$) and at fatigue (per protein, $P = 0.003$), whilst $[K^+]_a$, $[K^+]_v$ and $[K^+]_{a-v}$ were each increased at fatigue ($P = 0.001$). During FF, in DIG (main effects), time to fatigue, $[K^+]_a$, $[K^+]_v$ and $[K^+]_{a-v}$ were unchanged; with exercise (main effects), plasma $[K^+]_a$, $[K^+]_v$, $[K^+]_{a-v}$ and muscle $K^+$ efflux were all increased at fatigue ($P = 0.001$). Thus, muscle strength declined, but functional muscle NKA content was preserved during DIG, despite elevated plasma digoxin and muscle NKA–digoxin occupancy, with $K^+$ disturbances and fatiguability unchanged.

(Received 1 March 2022; accepted after revision 8 July 2022; first published online 15 July 2022)

**Corresponding author** Professor Michael J. McKenna: Institute for Health and Sport, Victoria University, PO Box 14428, Melbourne, Victoria 8001, Australia. Email: michael.mckenna@vu.edu.au

**Abstract figure legend** Digoxin specifically inhibits Na,K-pumps in all tissues and in skeletal muscle, and thus could therefore impair cellular Na/K homeostasis, excitability and contractility. In heart failure patients, digoxin binds to and therefore reduces the Na,K-pump content in skeletal muscle; this lower number of available functional Na,K-pumps is consistent with an elevated circulating [K] during exercise. We show here in healthy volunteers that oral digoxin intake, which resulted in therapeutic [digoxin], did not reduce the muscle Na,K-pump content, which was unchanged. However, measures with digibind revealed the total number of Na,K-pumps was elevated by 8%. Digoxin did not affect either arterial [K] or time to fatigue, during both finger flexion exercise and leg cycling exercise. This indicates a remarkable preservation of skeletal muscle Na,K-pumps and thus also of circulating [K] and performance during fatiguing, intense exercise challenges. However, one adverse consequence of digoxin was a 4% reduction in muscle strength.

## Key points

- The $Na^+,K^+$-ATPase (NKA) is vital in regulating skeletal muscle extracellular potassium concentration ($[K^+]$), excitability and plasma $[K^+]$ and thereby also in modulating fatigue during intense contractions.
- NKA is inhibited by digoxin, which in cardiac patients lowers muscle functional NKA content ($[^3$H]-ouabain binding) and exacerbates $K^+$ disturbances during exercise.
- In healthy adults, we found that digoxin at clinical levels surprisingly did not reduce functional muscle NKA content, whilst digoxin removal by Digibind antibody revealed an $\sim$8% increased muscle total NKA content.
- Accordingly, digoxin did not exacerbate arterial plasma $[K^+]$ disturbances or worsen fatigue during intense exercise, although quadriceps muscle strength was reduced.
- Thus, digoxin treatment in healthy participants elevated serum digoxin, but muscle functional NKA content was preserved, whilst $K^+$ disturbances and fatigue with intense exercise were unchanged. This resilience to digoxin NKA inhibition is consistent with the importance of NKA in preserving $K^+$ regulation and muscle function.

## Introduction

Intense exercise causes major disturbances in potassium concentration ($[K^+]$) in intracellular and extracellular fluids in blood and muscle. Intense cycling exercise reduced muscle intracellular $[K^+]$ by $\sim$22%, whilst muscle interstitial $[K^+]$ rose to $10-15$ mmol l$^{-1}$ during intense leg muscle contractions (Green et al., 2000; Juel, Pilegaard et al., 2000; Nielsen et al., 2004; Sjogaard et al., 1985) and these disturbances have been linked to impaired muscle membrane excitability, reduced maximal force and fatigue (Lindinger & Cairns, 2021; McKenna et al., 2008; Sejersted & Sjogaard, 2000). During intense cycling, running or rowing, $[K^+]$ in arterial plasma reaches $6-8$ mmol l$^{-1}$ and even higher in femoral venous plasma; this is followed by rapid post-exercise reductions in $[K^+]$, often to sub-resting values or even hypokalaemia (Atanasovska et al., 2014, 2018; Lindinger et al., 1992; McKenna, Heigenhauser, McKelvie, MacDougall et al., 1997; Medbo & Sejersted, 1990; Vollestad et al., 1994). Post-exercise hypokalaemia is also associated with impaired cardiac repolarisation and increased risk of arrhythmias (Atanasovska et al., 2018; Lindinger & Cairns, 2021; Tran et al., 2022). Hence, K$^+$ regulation is critical both during and after exercise.

In skeletal muscle, the Na$^+$,K$^+$-ATPase (NKA) is a vital regulator of Na$^+$ and K$^+$ gradients across sarcolemmal and transverse tubular membranes, directly modulating membrane potential and excitability and thereby also having a fundamental role in maintaining contractility and resisting fatigue (Clausen, 2013; Lindinger & Cairns, 2021; McKenna et al., 2008; Sejersted & Sjogaard, 2000). Any changes in muscle NKA *in vivo* may therefore modulate $[K^+]$ fluxes with intense exercise and muscle function (McKenna et al., 2008). Muscle NKA content, as fully quantified by the [$^3$H]-ouabain binding site content, is increased by $8-22$% by exercise training for days to months, across a range of exercise modalities and conditions; training also typically lowered plasma and/or muscle interstitial $[K^+]$ during exercise and enhanced exercise performance (Harmer et al., 2000; McKenna et al., 1993; Nielsen et al., 2004; Wyckelsma et al., 2019). Increased muscle NKA content by dexamethasone also

reduced plasma $[K^+]$ and muscle K$^+$ release during exercise and tended to enhance exercise performance (Nordsborg et al., 2005, 2008). Conversely, reduced muscle NKA content and muscle strength occurred in patients with severe inactivity due to knee injury (Perry et al., 2015), whilst inactivity via lower limb suspension failed to lower NKA content, but reduced strength and exercise performance (Perry et al., 2016).

Despite NKA's importance in maintaining muscle function against ouabain inhibition in animal isolated muscle preparations and in gene manipulation models (Clausen, 2013; Nielsen & Clausen, 1996; Radzyukevich et al., 2013), it remains difficult to extrapolate these findings to *in vivo* functional effects of NKA in contracting human skeletal muscle. However, clinical use of the cardiotonic steroid digoxin allows exploration of *in vivo* effects of specific NKA inhibition. Digoxin is used to treat patients with severe heart failure and atrial fibrillation (Angraal et al., 2019; Bavendiek et al., 2017). Digoxin exerts a positive myocardial inotropic effect, largely via inhibition of NKA activity causing increased myocardial intracellular Na$^+$, Na$^+$/Ca$^{2+}$ exchange and cytosolic Ca$^{2+}$ (Levi et al., 1994); digoxin also exerts myocardial chronotropic and dromotropic effects via neurohumoral mechanisms (Maury et al., 2014). Circulating digoxin also binds to and inhibits a fraction of the far larger NKA pool in skeletal muscle, which is $\sim$50% of total body digoxin binding (Schmidt & Kjeldsen, 1991; Schmidt et al., 1991). Heart failure patients undergoing digitalisation have a digoxin occupancy of NKA binding sites in skeletal muscle of $\sim$9% after 3 days of treatment (Schmidt et al., 1995) and 13% in muscle obtained at post-mortem (Schmidt, Holm-Nielsen et al., 1993), with corresponding reductions in the [$^3$H]-ouabain binding site content in myocardium and muscle (Schmidt et al., 1991, 1995; Schmidt, Allen et al., 1993). Hence, a deficit persists in muscle functional NKA content in digitalised heart failure patients. Consistent with this, digoxin also exacerbated $[K^+]$ increases with exercise. In patients with atrial fibrillation, a serum digoxin concentration ([digoxin]) of $1-2$ nmol l$^{-1}$ was associated with a 1 mmol l$^{-1}$ greater rise in serum $[K^+]$ during exercise (Norgaard et al., 1991). In heart failure patients, 1 nmol l$^{-1}$

**Simon Sostaric** completed his PhD at Victoria University, Melbourne, Australia, investigating the effects of alkalosis and digoxin on plasma potassium, ionic homeostasis and exercise performance in healthy humans. His current research interests encompass exercise-induced muscle fatigue, thermoregulation and metabolic disturbances. He is the founder of Melbourne Sports & Allied Health Clinic, and practices as a clinical exercise physiologist, with a focus on metabolic and cardiovascular conditions. **Aaron Petersen** completed his PhD at Victoria University, Melbourne, Australia, where he investigated the roles of potassium and sodium–potassium pumps on muscle fatigue. Following a post-doctoral fellowship at Deakin University, he returned to Victoria University, where he is an Associate Professor of Exercise Physiology. His research focuses on investigating interventions that can maintain or enhance physical function in both healthy and clinical populations, with a particular emphasis on environmental stressors.

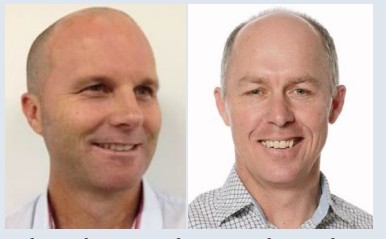

[digoxin] was associated with 0.2–0.3 mmol l$^{-1}$ higher increases in plasma [K$^+$] during intense exercise and 138% increased K$^+$ loss from contracting muscles (Schmidt et al., 1995). In healthy humans, high-dose digoxin (0.37–0.5 mg day$^{-1}$, 10 ds) increased resting venous [K$^+$] (Edner et al., 1993) and (0.5 mg day$^{-1}$, 10 days) increased digoxin binding to muscle and reduced serum [digoxin] during acute exercise (Joreteg & Jogestrand, 1983). Whether digoxin lowers muscle NKA content in healthy individuals at rest and during exercise and thereby also impairs K$^+$ regulation and affects fatigue with intense exercise are unknown.

We therefore explored the effects of 14 days of oral digoxin in healthy adults, in a carefully controlled, integrated experiment designed to simulate typical clinical use of digoxin, on each of muscle NKA content, K$^+$ regulation and fatigue with intense exercise models utilising a large (two-legged cycling) and a small (finger flexion) muscle mass and fatigue also during repeated quadriceps contractions. The following hypotheses were tested. That digoxin would (i) increase fatiguability during all three exercise modes and for skeletal muscle NKA, would: (ii) bind to and inhibit NKA, measured by lowered [$^3$H]-ouabain binding; for two-legged cycling, that digoxin would (iii) increase arterial [K$^+$] during and after exercise and (iv) decrease K$^+$ uptake by inactive forearm muscle; and for finger flexion exercise, that digoxin would: (v) exacerbate increases in arterial and antecubital venous [K$^+$] and (vi) increase forearm muscle K$^+$ release during exercise and reduce K$^+$ re-uptake following exercise.

## Methods

### Ethical approval

Ethical clearance was obtained from the Victoria University Human Research Ethics Committee and the Alfred Hospital Ethics Committee and conformed to the standards set by the *Declaration of Helsinki*, except for registration in a database.

### Participants

The study was undertaken in healthy young adults to avoid possible confounding pharmacological, disease or inactivity-related effects that might exist if studying digoxin effects in clinical populations. Ten healthy, untrained but recreationally active individuals, comprising nine males and one female, gave written informed consent and participated in the study [age 26.1 (5.9) years, mass 75.7 (11.3) kg, and height 178.4 (9.2) cm; V$_{O_2 peak}$, 3.67 (0.42) l min$^{-1}$; mean (SD)]. All participants underwent an initial medical examination to screen for abnormal plasma electrolyte concentrations, kidney function, rest and exercise ECG, and history of adverse cardiovascular events. Participants consumed pre-packaged iso-energetic meals and beverages during 72 h before each experimental trial and refrained from vigorous activity and ingestion of caffeine and alcohol in the 24 h before each visit.

### Study design and digoxin administration

After initial medical screening, participants attended the exercise physiology laboratory on six separate occasions. During their initial laboratory visit, anthropometric measurements of forearm length and circumferences were made for forearm volume estimation and participants were then familiarised with finger flexion contractions (Sostaric et al., 2006). Participants performed an initial incremental finger flexion exercise test and, after 30 min rest, were familiarised with the high-intensity, intermittent finger flexion protocol for use in subsequent trials. After a further 10 min rest, participants then performed an incremental two-legged cycling test to determine peak oxygen consumption. Participants repeated the finger flexion exercise test during the second and third laboratory visits, to determine intra-subject variability in power output and time to fatigue. Participants were also familiarised with the leg cycling exercise test protocol for use in subsequent trials, and also underwent two variability trials on separate days.

Participants were allocated to either a digoxin (DIG) or placebo (CON) initial treatment group, with trials conducted using a randomised, double-blind, crossover, counterbalanced design. Participants ingested an oral dose of digoxin 0.25 mg day$^{-1}$ (Lanoxin; GlaxoSmithKline, Victoria, Australia) or a placebo (sugar tablet) for 14 days. This dosage and delivery simulated typical clinical usage of digoxin and allowed sufficient time for [digoxin] to reach steady state concentrations. After a 4 week washout period, participants undertook the alternative treatment for a further 2 weeks and then repeated the experimental exercise tests. Thus, for those participants taking DIG first, the effective digoxin clearance time was 6 weeks, that is the standard 4 week washout plus the 2 weeks of placebo treatment. Similarly, for the placebo group, the time between biopsies was 6 weeks. This washout period should be more than sufficient, given that digoxin clearance half-time from serum after digoxin injection was ∼45 h (range 32−131 h) (Kramer et al., 1974) and from skeletal muscle after oral digoxin was ∼2.2 days (Jogestrand & Sundqvist, 1981).

Participants underwent quadriceps muscle function testing on day 13 of treatment, and on day 14 performed finger flexion (FF) followed by leg cycling (LC) exercise tests, with repeated arterial and venous blood sampling.

Participants rested for 3 h between FF and LC trials, with arterial and venous [K$^+$] analysed to ensure resting values had been re-established before commencement of the LC trial. An additional venous blood sample was taken at rest to measure serum [DIG] on days 7, 13 and 14 of the treatment period. Heart rate and rhythm were monitored during experimental exercise tests by a 12-lead ECG (Mortara, Boston, MA, USA). For ethical and safety reasons, the attending medical practitioner on day 14 trials was non-blinded; also, to minimise clinical risks, all digoxin measurements were performed by the Alfred Hospital for rapid reporting of [digoxin]. Study protocols are shown schematically in Fig. 1*A*.

### Exercise protocols

**Quadriceps muscle strength and fatiguability test.** Participants performed tests of quadriceps muscle strength and endurance (dominant leg) on an isokinetic dynamometer (Cybex Norm 770, Henley Healthcare, USA) with all test protocols as previously detailed (Li et al., 2002). Three familiarisation trials were conducted at least 2 weeks before commencement of the trial, to minimise any training effect. The dynamometer was calibrated for angle, torque and velocity immediately before each test. Participants initially warmed up by cycling for 3 min at 50 W. They were then strapped to the Cybex chair with belts across the hips, chest and leg to stabilize the upper body and thigh. To minimise variability the same positions were recorded for each individual and used in all trials for that person. At each contraction velocity, after submaximal familiarisation contractions, participants performed two practice maximal repetitions,

had 1 min of rest and then performed three maximal dynamic repetitions, each separated by 2 min rest. Maximal peak torque was measured at 0, 60, 120, 180, 240, 300 and 360 s$^{-1}$ and the peak torque from the three contractions at each velocity was used to construct a torque–velocity relationship. After 10 min rest, participants then performed the quadriceps muscle fatigue test. Fatigue was determined from the percentage decline in peak torque during 50 repeated maximal contractions, conducted at 180 s$^{-1}$, at 0.5 Hz, with intervening passive knee flexion. Peak torque was defined as the average of the five highest peak torque values of the first 10 contractions, while final torque was the average of the five highest values of the last 10 contractions. This conservative approach minimised impacts of artificially low initial, as well as final, contractions. Fatigue was expressed as a fatigue index: Fatigue Index (%) = ((Peak torque − Final torque)/Peak torque) × 100.

**Finger flexion exercise test.** All finger flexion exercise tests were conducted on a custom designed dynamometer, with participants generating force against resistance, at 30 contractions min$^{-1}$, with fatigue defined as a failure to maintain power output and/or cadence for eight consecutive contractions (Sostaric et al., 2006). During the initial incremental finger flexion exercise test, contractions commenced at mean power output of 3.37 (0.60) W, with resistance and thus power output increased at the end of each minute, such that power output increased by 0.25 (0.15) W each minute. Contractions continued until volitional fatigue to allow determination of their peak work rate (WR$_{peak}$) and calculation of the work rate corresponding to 105% WR$_{peak}$ for use in all subsequent

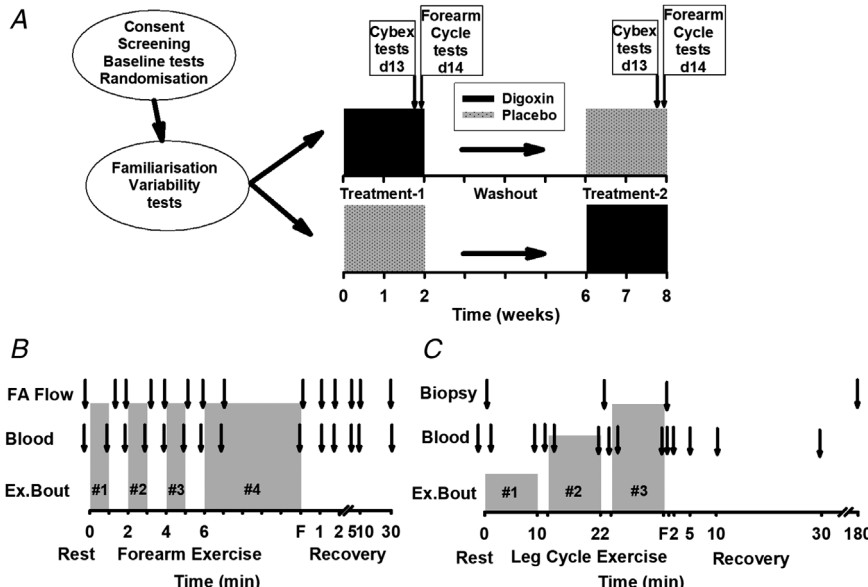

**Figure 1. Protocol overview for testing, digoxin, forearm exercise and leg cycle exercise**
*A*, after giving informed consent, participants underwent screening, baseline testing and were randomised into treatment groups initially taking either digoxin or placebo, and in a crossover design, followed then by the alternative treatment; participants then underwent familiarisation and variability testing, with all test procedures and timing detailed in the methods. *B*, protocol for forearm exercise trials (FA) indicating timing of forearm blood flow measures and of arterial and antecubital venous blood sampling, in relation to exercise and recovery times. *C*, protocol for leg cycling exercise trials indicating timing of muscle biopsies and of arterial and antecubital venous blood sampling, in relation to exercise and recovery times. F, fatigue.

trials. Participants performed a high-intensity, intermittent finger flexion exercise test (FF) that was identical during each of the two variability trials and the two experimental trials. The FF protocol comprised three exercise bouts (EB1–EB3) of 1 min duration interspersed by 1 min rest, followed by a fourth and final bout continued to fatigue (EB4), with all bouts conducted at a work rate corresponding to 105% $WR_{peak}$.

**Incremental leg cycling exercise test.** Participants performed an initial incremental cycling test to determine peak oxygen uptake ($V_{O_2peak}$) and to calculate work rates corresponding to 33, 67 and 90% $V_{O_2peak}$ for subsequent trials. The $V_{O_2peak}$ trial comprised 4 min each at 60, 90, 120 and 150 W, then 25 W increments per minute continued until volitional fatigue, with all equipment, calibration methods and procedures as previously described (Li et al., 2002).

The $V_{O_2peak}$ test and all subsequent leg cycling tests were conducted on an electrically braked cycle ergometer (Lode Excalibur, Groningen, Holland), pedalling at 70−80 rpm, with participants seated in a modified partially recumbent position, using a custom-designed seat with back support, with their dominant arm secured and elevated at heart level, and thus rendered inactive. This set-up was used to facilitate blood and muscle sampling during the experimental exercise trials.

**Experimental leg cycling test.** Participants performed a leg cycling exercise test (LC) that was identical during each of the two variability trials and the two experimental trials. The protocol comprised cycling for 10 min each at work rates corresponding to 33% $V_{O_2peak}$ (EB1) and 67% $V_{O_2peak}$ (EB2), each followed by 2 min rest and then a third and final bout continued to fatigue at a work rate corresponding to 90% $V_{O_2peak}$ (EB3). Fatigue for all trials was defined as a failure to maintain pedal cadence above 55 rpm for 10 consecutive seconds.

**Rationale for the finger flexion and leg cycling exercise test models.** Participants undertook two exercise models to contrast in the same individual the effects of a markedly different active muscle mass on disturbances of $K^+$ and other ions, during and following exercise. We utilised different exercise models and both arterial and venous blood sampling sites, as these each potentially affect $K^+$ regulation. Whilst continuous exercise typically induces intensity-dependent, sustained or continuous increases in arterial plasma $[K^+]$, intermittent exercise induces sharp oscillations in arterial $[K^+]$, with the magnitude of changes dependent on the intensity and duration of bouts, as well as the recovery interval (Altarawneh et al., 2016; Atanasovska et al., 2014, 2018; Lindinger et al., 1992; Vollestad et al., 1994). Since the rapid decline in $[K^+]$ immediately following each intermittent exercise bout is an indicator of *in vivo* NKA activation, intermittent exercise bouts were used in both exercise models to investigate digoxin effects on $[K^+]$ regulation. Intense finger flexion exercise, using a small contracting muscle mass, was utilised because this induced sharp elevations in $[K^+]$ in antecubital venous blood draining contracting forearm muscles, but barely perturbed arterial concentrations, reflecting $K^+$ release from contracting muscles (Sostaric et al., 2006). In contrast, intense two-legged cycling exercise, utilising a large contracting muscle mass, was utilised because this induces marked increases in arterial and femoral venous plasma $[K^+]$, but lesser increases in antecubital venous plasma, reflecting $K^+$ uptake by the inactive forearm muscles (Kowalchuk, Heigenhauser, Lindinger, Obminski et al., 1988; Kowalchuk, Heigenhauser, Lindinger, Sutton et al., 1988; Lindinger et al., 1990, 1992, 1995; McKenna, Heigenhauser, McKelvie, MacDougall et al., 1997; McKenna, Heigenhauser, McKelvie, Obminski et al., 1997).

### Blood sampling procedures and analyses

Catheters (20 or 22G; Jelco) were inserted anterograde in the radial artery (a) of the non-contracting arm, under local anaesthesia (2% lignocaine injection) and retrograde into the deep antecubital vein (v) of the contracting forearm. Participants then rested for ∼30 min before the commencement of each DIG and CON trial. Intra-arterial and intravenous blood pressures were continually monitored (Marquette 710, Wisconsin, USA) using electronic pressure transducers (Abbott Critical care Systems, Chicago, IL, USA) connected to the saline-filled cannula via an extension line. Blood pressure signals were then interfaced with the finger flexion exercise computer system, enabling continual integration between power output, forearm circumference changes and blood pressure data. Arterial lines were kept patent by a slow, sterile, isotonic saline infusion.

During FF trials, arterial and venous blood samples (5 ml) were taken simultaneously at rest, in the final 10 s of each of the three initial 1 min exercise bouts (EB1, EB2, EB3), immediately before the start of each subsequent exercise bout (PreEB2, PreEB3 and PreEB4), during the final exercise bout at both 1 min (EB4+1) and during the final contractions at fatigue, as well as at 1, 2, 5, 10 and 30 min following exercise. Hand blood flow was occluded for 10 s before and during venous blood sampling by a high-pressure wrist cuff, to exclude collection of hand venous effluent blood (Fig. 1*B*). Forearm blood flow was determined using plethysmography and conducted immediately following blood sampling at rest, during exercise and in recovery, as previously described (Sostaric et al., 2006).

During LC trials, arterial and venous blood samples (5 ml) were taken simultaneously at rest, in the final 10 s of each of the two initial exercise bouts (EB1, EB2), immediately before the start of each subsequent exercise bout (PreEB2, PreEB3), during the final exercise bout at both 1 min (EB3+1) and at fatigue, and at 1, 2, 5, 10 and 30 min following exercise (Fig. 1*C*). Hand blood flow was occluded for 10 s before and during venous blood sampling by a high-pressure wrist cuff, to exclude collection of hand venous effluent blood. Forearm blood flow was determined using plethysmography and conducted immediately following blood sampling at rest, during exercise and in recovery, but the data were discarded as unreliable, being affected by movement artefacts.

For measurement of serum [digoxin], blood samples were allowed to clot and then centrifuged at 4000 rpm for 10 min at 4°C (3K15 refrigerated centrifuge; Sigma, Laborzentrifugen, Germany). Serum was subsequently stored at −20°C until assayed for serum digoxin by standard clinical analysis using a Multigent (Abbot Laboratories, IL, USA) homogenous particle-enhanced turbidimetric immunoassay. The serum [digoxin] minimum detection level reported by the Alfred Hospital was 0.4 nmol l$^{-1}$ for the first seven participants, but changed during the study to 0.2 nmol l$^{-1}$ for the final three participants. However, this is not an important limitation given the clear elevation of [digoxin] to expected levels in the digoxin trials.

Arterial and venous blood samples from the experimental trials were immediately analysed in duplicate for plasma [K$^+$] using an automated blood gas analyser (Sostaric et al., 2006). Whole blood [Hb], Hct and SO$_2$, and plasma $P_{O_2}$ and $P_{CO_2}$ were analysed (data not reported) using automated haematology and blood gas analysers (Sostaric et al., 2006). All measurements and analysers were calibrated immediately before, during and after measurements with precision standards in the range of the measurements (Sostaric et al., 2006).

**Calculations for plasma and blood.** Changes from resting levels in plasma volume ($\Delta PV_a$) and blood volume ($\Delta BV_a$) and changes in venous compared to arterial plasma ($\Delta PV_{a-v}$) and blood volume ($\Delta BV_{a-v}$) across the forearm were calculated during and following exercise, from changes in [Hb] and Hct (data not reported), as previously described (McKenna, Heigenhauser, McKelvie, MacDougall et al., 1997; Sostaric et al., 2006). These calculations enabled corrections to be made for the effects of fluid shifts on K$^+$ in plasma and blood during and following exercise. Plasma K$^+$ efflux data were corrected for fluid shifts. Plasma [K$^+$]$_{a-v}$ (mmol l$^{-1}$) was corrected for $\Delta PV_{a-v}$ using the equation:
$$[ion]_{a-v} = ([ion]_a/(1 + \Delta PV_{a-v})) - [ion]_v$$ (McKenna,

Heigenhauser, McKelvie, MacDougall et al., 1997). Net K$^+$ fluxes across the forearm were calculated as the product of corrected [K$^+$]$_{a-v}$ and plasma blood flow, and expressed in $\mu$mol min$^{-1}$. Plasma flow was calculated as forearm blood flow × Hct (expressed as a fraction) (Sostaric et al., 2006).

## Muscle sampling and analyses

**Muscle biopsy procedures.** A local anaesthetic (1% lignocaine) was injected into the skin and subcutaneous tissue over the vastus lateralis muscle, four small incisions (two per leg) were made through the skin and fascia, and a muscle sample of ∼100−120 mg was then excised using a biopsy needle. All biopsies were taken during the LC trial on day 14 of both DIG and CON trials. A biopsy was taken at rest, immediately after exercise at 67% $V_{O_2peak}$, after 90% $V_{O_2peak}$ to fatigue, and at 3 h following exercise, giving a total of four biopsies per trial and eight in total. The exercise and recovery biopsies were included to detect possible digoxin effects on NKA–digoxin binding influenced by exercise. Samples were immediately frozen in liquid N$_2$ until assayed later for NKA [$^3$H]-ouabain binding site content and maximal *in vitro* K$^+$-stimulated 3-*O*-MFPase activity.

**[$^3$H]-ouabain binding site content.** Skeletal muscle NKA content was determined in quadruplicate by measurement of vanadate-facilitated [$^3$H]-ouabain binding site content (OB), using standard techniques (Nørgaard et al., 1983, 1984), as previously described for human muscle (Petersen et al., 2005). The OB assay was performed without previous incubation in digoxin antibody fragments (F$_{ab}$) and is therefore designated hereafter as OB-F$_{ab}$. The muscle protein content was determined spectrophotometrically as described (Petersen et al., 2005) and the OB-F$_{ab}$ expressed as pmol (g wet weight)$^{-1}$ and pmol (g protein)$^{-1}$. The intra- and inter-assay CV for the OB assay were 9% and 14%, respectively.

**Digoxin binding and occupancy.** As digoxin and ouabain bind competitively to the digitalis receptors on the NKA, the OB assay will not detect any NKA already occupied by digoxin. Thus, the OB assay was also performed in separate muscle pieces after previous incubation in digoxin F$_{ab}$ (Digibind; GlaxoSmithKline), using previously described procedures (Schmidt & Kjeldsen, 1991; Schmidt, Holm-Nielsen et al., 1993), which were demonstrated to remove ∼97% of bound digoxin from skeletal muscle (Schmidt & Kjeldsen, 1991). Briefly, muscle samples were incubated for 16 h at 30°C in buffer containing 250 mм sucrose, 10 mм Tris, 3 mм MgSO$_4$, 1 mм NaVO$_4$ and 0.5 $\mu$M F$_{ab}$ (pH 7.2–7.4), after which

the standard OB assay was performed. This measure of OB is therefore designated hereafter as OB+$F_{ab}$. The skeletal muscle digoxin occupancy was calculated for each individual as the difference in OB with $F_{ab}$ incubation (OB+$F_{ab}$) minus the standard OB without $F_{ab}$ incubation (OB-$F_{ab}$), expressed as a proportion of total OB sites: [Occupancy = (OB+$F_{ab}$ − OB-$F_{ab}$) × 100/(OB+$F_{ab}$)] (Schmidt, Holm-Nielsen et al., 1993). This calculation was performed for each of the DIG and CON trials.

**$F_{ab}$ control experiments.** Two control experiments were conducted to test the efficacy of the digoxin $F_{ab}$ method in detecting bound NKA in our laboratory. These were to verify that use of $F_{ab}$ would enable recovery of bound NKA and that the 16 h incubation would not affect OB in healthy human skeletal muscle.

*Recovery.* Vastus lateralis muscle samples from three healthy participants were each divided into three portions, for incubation in digoxin, digoxin+$F_{ab}$ and control ($n = 3$ per group). The digoxin and digoxin+$F_{ab}$ samples were incubated in 10 nM digoxin for 60 min at 37°C. The digoxin+$F_{ab}$ samples were subsequently incubated overnight in 0.5 $\mu$M digoxin $F_{ab}$, whilst the control samples did not undergo any incubation. The digoxin, digoxin+$F_{ab}$ and control samples were then analysed for OB as described above. The clearance of bound digoxin was calculated from the OB results of the digoxin, digoxin+$F_{ab}$ and control samples, using the formula: [(digoxin + $F_{ab}$ − digoxin)/(control − digoxin)] × 100 (Schmidt & Kjeldsen, 1991). Incubation of human muscle in 10 nM digoxin [mean(SD) 276(83) pmol g wet weight$^{-1}$] reduced OB by 32% compared to control [406(19) pmol g wet weight$^{-1}$, $P = 0.021$], whereas following incubation in $F_{ab}$ [401(11) pmol g wet weight$^{-1}$], the OB did not differ from control ($P = 0.918$). Thus, the $F_{ab}$ removed 96% of bound digoxin (i.e. 96% of the difference between control and digoxin) and enabled almost complete NKA quantification, similar to previous reports (Schmidt & Kjeldsen, 1991).

*Incubation effects.* To ensure that the additional 16 h overnight incubation with $F_{ab}$ did not affect OB, an additional three human vastus lateralis muscle samples from different participants were tested. To compare the effect of incubation in $F_{ab}$ *vs.* incubation in standard TVS buffer, samples were incubated for 16 h at 30°C in standard TVS buffer or buffer containing 0.5 $\mu$M $F_{ab}$ ($n = 3$ per group). To determine if the 16 h incubation in $F_{ab}$ or TVS buffer affected subsequent [$^{3}$H]-ouabain binding, an additional three samples did not undergo any incubation (control). The TVS buffer, $F_{ab}$ and control samples were then analysed for OB. Overnight incubation in standard TVS buffer [TVS, mean(SD), 320(36) pmol g wet weight$^{-1}$] or in buffer containing $F_{ab}$ [Digibind, 310(26) pmol g wet weight$^{-1}$] had no effect on OB

[Control, 315(35) pmol g wet weight$^{-1}$] ($P = 0.929$), indicating that the overnight incubation did not result in any loss of NKA, also similar to previous reports (Schmidt & Kjeldsen, 1991).

**Muscle 3-*O*-MFPase assay.** The maximal *in vitro* K$^{+}$-stimulated 3-*O*-methylfluorescein phosphatase (3-*O*-MFPase) activity assay adapted for human skeletal muscle was measured in muscle homogenates as a marker of NKA in resting muscle, with NKA specificity demonstrated by being ouabain-inhibitable (Fraser & McKenna, 1998). Numerous previous studies have reported reduction in the maximal *in vitro* 3-*O*-MFPase activity with fatiguing exercise, suggesting this reflected a corresponding reduction in maximal NKA activity (McKenna et al., 2008). Others have argued that the maximal *in vitro* 3-*O*-MFPase activity cannot validly measure changes with exercise in the *in vitro* NKA activity, because the Na$^{+}$-dependent NKA activity directly measured via $^{33}$Pi liberation from $^{33}$P-ATP hydrolysis in a fractionated preparation was not reduced after heavy exercise, whereas the (Na$^{+}$-insensitive) maximal 3-*O*-MFPase activity was reduced (Juel et al., 2013). However, the same group also did report a reduction in the maximal Na$^{+}$-dependent, ATPase activity following exercise (Hostrup et al., 2014). These findings (Juel et al., 2013) do not challenge whether intervention-induced changes in the 3-*O*-MFPase activity in resting muscle can reflect changes in maximal NKA activity, where intracellular [Na$^{+}$] is low, but rather, query their significance in exercised muscle, where [Na$^{+}$] is elevated (McKenna et al., 2008). Given the conflicting findings from that group on exercise effects on muscle NKA activity (Hostrup et al., 2014; Juel et al., 2013), here we also report the 3-*O*-MFPase activity measure as a marker of NKA activity in exercised muscle. Approximately 20 mg of muscle was analysed, with all assays performed at 37°C, with continuous stirring, on a fluorometer (Photon Technology International, Birmingham, NJ, USA). The 3-*O*-MFPase maximal *in vitro* activity results were expressed relative to wet weight, and also relative to muscle protein content, to correct for any exercise-induced changes in muscle water content. The intra-assay CV for the 3-*O*-MFPase assay was 18%.

### Statistical analyses

Results are expressed as mean (SD). Data sets were tested for normality using the Shapiro–Wilk test and if criteria were not met, data were log transformed. Data sets were analysed using a linear mixed model, with time (rest, exercise, recovery) and treatment (DIG, CON) as fixed effects, and restricted maximum likelihood as the estimation method for any missing values. *Post hoc*

analyses used the Least Significant Difference test. Where significant time effects were found, differences between times are indicated, but non-significant differences between times are not detailed. Significant treatment effects are reported (i.e. DIG *vs.* CON). To avoid repetition, treatment-by-time interactions are only stated when significant.

For the muscle data, a one-way ANOVA was used for the digoxin F$_{ab}$ verification experiments. Paired data in resting muscle were analysed using a paired Student *t* test, for comparison of OB+F$_{ab}$ *vs.* OB-F$_{ab}$. Sample size was balanced ($n = 10$) for all OB ($\pm$F$_{ab}$) and associated digoxin occupancy measures, and all 3-*O*-MFPase activity measures. The exact number of measurement points, defined as number of participants $\times$ number of trials $\times$ number of measurements minus missing data points, was 76, 76 and 77 for OB-F$_{ab}$, OB+F$_{ab}$ and 3-*O*-MFPase activity, respectively.

Sample size for functional tests was $n = 10$ for Cybex tests and leg cycling exercise performance and $n = 9$ for finger flexion exercise performance. The exact number of data measurement points for Cybex tests was 138 for torque–velocity and 1000 for fatigue data and for time to fatigue during FF was 18 and during LC was 20. For blood analyses with exercise, balanced data sets between trials were used, with all data for a participant discarded if their alternative trial data were unavailable; sample sizes therefore differed between variables. During FF, sample sizes for arterial analyses were $n = 9$ for all variables. For FF venous and [ion]$_{a\text{-}v\text{ differences}}$ (uncorrected), sample sizes were $n = 8$ for [K$^+$] and $n = 7$ for all other variables. For FF [ion]$_{a\text{-}v\text{ difference}}$ (corrected for fluid shifts), sample sizes were $n = 7$ for [K$^+$]; whilst for plasma [ion] fluxes, $n = 7$ for [K$^+$]. The exact number of measurement points during FF for [K$^+$]$_a$, [K$^+$]$_v$, [K$^+$]$_{a\text{-}v}$ was 242, 218 and 201, respectively. For calculated K$^+$ flux data during FF, the exact number of data measurement points was 168. For LC, sample sizes for arterial analyses were $n = 10$ for all variables. For LC venous and [ion]$_{a\text{-}v\text{ difference}}$ (uncorrected), sample sizes were $n = 8$ for all variables. For LC [ion]$_{a\text{-}v\text{ difference}}$ (corrected for fluid shifts), sample sizes were $n = 8$ for [K$^+$]. The exact number of measurement points during LC for [K$^+$]$_a$, [K$^+$]$_v$ and [K$^+$]$_{a\text{-}v}$, was 218, 216 and 209, respectively. Paired data were analysed using a paired Student *t* test (e.g. exercise time to fatigue). For paired comparisons, $n = 10$, except for FF where $n = 9$.

Individual variability was calculated for all participants within the exercise protocol ($n = 10$) and averaged to obtain an overall coefficient of variation (CV), with method variability also determined by CV. Statistical significance was accepted at $P < 0.05$. Statistical analyses were performed using IBM SPSS Statistics 27; since this package does not report exact *P* values lower than $P < 0.001$, for these data they are reported here as $P = 0.001$.

## Results

### Serum digoxin concentration and compliance

All participants reported full compliance with digoxin, and no adverse events were reported. During DIG, serum [digoxin] on days 7, 13 and 14 were 0.7 (0.2), 0.7 (0.2) and 0.8 (0.2) nmol l$^{-1}$, respectively [mean (SD)]. During CON, serum [digoxin] was reported below the method detection limits in nine of 10 participants ($n = 6$, $<0.4$ nmol l$^{-1}$; $n = 3$, $<0.2$ nmol l$^{-1}$) and at the reported detection level (0.4 nmol l$^{-1}$) in one individual.

### Muscle NKA results

#### Muscle total protein content

*Exercise (time effect).*    Protein content was less than rest, after exercise at 67% V$_{O_2\text{peak}}$ ($-8\%$, $P = 0.007$) and at fatigue ($-9\%$, $P = 0.002$), but did not differ from rest at 3 h after exercise ($P = 0.368$) [rest, 0.198 (0.025); 67% VO$_{2\text{peak}}$, 0.182 (0.029); fatigue, 0.181 (0.022); and 3 h after exercise, 0.203 (0.023) g protein g muscle wet weight$^{-1}$].

*Digoxin (treatment effect).*    Protein content did not differ between DIG and CON ($P = 0.430$).

#### Muscle [³H]-ouabain binding site content without Digibind (OB-F$_{ab}$)

*Exercise.*    The [³H]-ouabain binding site content measured without incubation in Digibind (OB-F$_{ab}$) (pmol g wet weight$^{-1}$) was elevated above rest after exercise at 67% V$_{O_2\text{peak}}$ (10%, $P = 0.005$) but not at fatigue (5%, $P = 0.163$); the OB-F$_{ab}$ at 67% V$_{O_2\text{peak}}$ was also higher than at 3 h after exercise (12%, $P = 0.005$, Fig. 2*A*). The OB-F$_{ab}$ (pmol g protein$^{-1}$) showed a similar pattern, being increased at both 67% V$_{O_2\text{peak}}$ (21%, $P = 0.001$) and at fatigue (16%, $P = 0.003$); these were each respectively greater than at 3 h recovery (67% V$_{O_2\text{peak}}$: 20%, $P = 0.001$; and fatigue: 16%, $P = 0.002$, Fig. 2*B*).

*Digoxin.*    Neither the OB-F$_{ab}$ (pmol g wet weight$^{-1}$) ($P = 0.253$) nor the OB-F$_{ab}$ (pmol g protein$^{-1}$) ($P = 0.087$, $-5.6\%$) differed significantly between DIG and CON (Fig. 2).

#### Muscle [³H]-ouabain binding site content measured after clearance of bound digoxin by F$_{ab}$ (OB+F$_{ab}$)

*Exercise.*    No exercise effect was found for the OB+F$_{ab}$ (pmol g wet weight$^{-1}$, $P = 0.814$, Fig. 3*A*), but the OB+F$_{ab}$ expressed per gram of protein was higher at fatigue than at rest (11%, $P = 0.022$) and tended also to be higher at 67% V$_{O_2\text{peak}}$ (9%, $P = 0.087$); OB+F$_{ab}$ at 3 h after exercise was less than both 67% V$_{O_2\text{peak}}$ ($-13\%$, $P = 0.041$) and fatigue ($-15\%$, $P = 0.009$, Fig. 3*B*).

*Digoxin.* The OB+$F_{ab}$ did not differ between DIG and CON whether expressed per gram wet weight ($P = 0.809$) or per gram protein ($P = 0.376$, Fig. 3).

**Digoxin occupancy of muscle NKA.** In DIG, the OB+$F_{ab}$ in resting muscle was 8.2% higher than OB-$F_{ab}$ [381 (53) *vs.* 352 (54) pmol g wet weight$^{-1}$, respectively, $P = 0.047$]. The digoxin occupancy was then calculated as

the difference between OB+$F_{ab}$ and OB-$F_{ab}$ and expressed as a percentage of the total NKA (OB+$F_{ab}$), which in DIG was 7.6% of the total NKA (muscle OB+$F_{ab}$). In contrast, in CON, the OB+$F_{ab}$ in rest muscle did not differ from OB-$F_{ab}$ [369 (41) *vs.* 353 (42) pmol g wet weight$^{-1}$, respectively, 4.5%, $P = 0.20$]; the equivalent calculation of 'apparent occupancy' in CON was 4.3% of total NKA (muscle OB+$F_{ab}$).

**Maximal *in vitro* 3-*O*-MFPase activity**

*Exercise.* The 3-*O*-MFPase activity (nmol min$^{-1}$ g wet weight$^{-1}$) did not differ from rest at 67% $V_{O_2peak}$ but

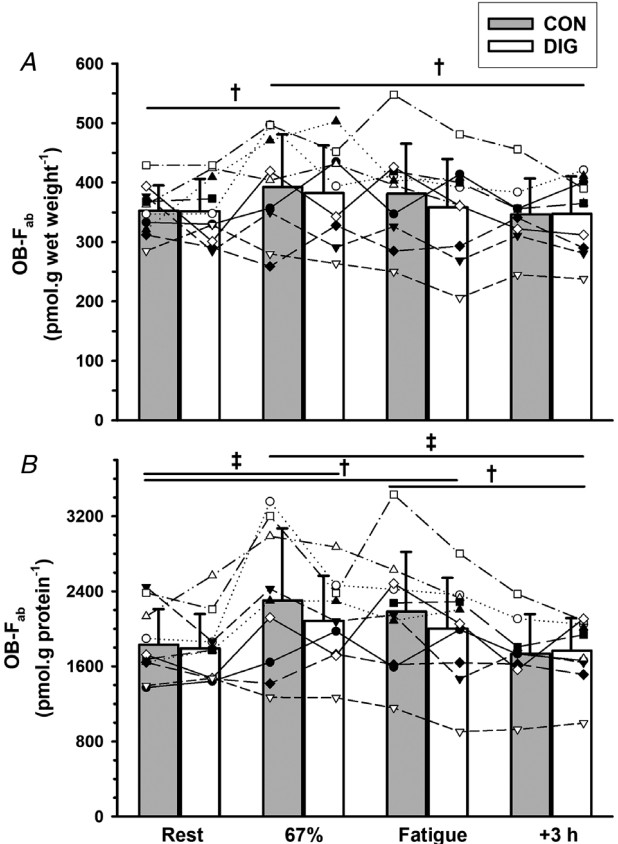

**Figure 2. Effects of digoxin and acute exercise on [³H]-ouabain binding site content in human skeletal muscle, without prior Digibind incubation (OB-$F_{ab}$)**
The [³H]-ouabain binding site content measured using the standard assay without Digibind incubation (OB-$F_{ab}$) after 14 days of oral digoxin (DIG) or placebo (CON), expressed relative to (*A*) muscle wet weight and (*B*) muscle protein content. Biopsies were taken at rest (Rest), after cycling for 10 min at 33% $V_{O_2peak}$ and 10 min at 67% $V_{O_2peak}$ (67%), immediately following cycling to fatigue at 90% $V_{O_2peak}$ (Fatigue) and at 3 h after exercise (+3 h). Data are mean (SD), *n* = 10. Exact number of data measurement points, defined as the number of participants × number of trials × number of measurements minus missing data points, was 76 for OB-$F_{ab}$. Individual data are plotted using different symbols and connected by dashed lines. Time effects are indicated as differences between the two times connected by a horizontal bar, †$P < 0.01$, ‡$P < 0.001$, with exact *P* values indicated in the Results. No significant digoxin effect was found for OB-$F_{ab}$ (per g wet weight, $P = 0.253$) or OB-$F_{ab}$ (per g protein, $P = 0.087$). Time × treatment interactions were not significant ($P = 0.851$, $P = 0.672$, per g wet weight or per g protein, respectively).

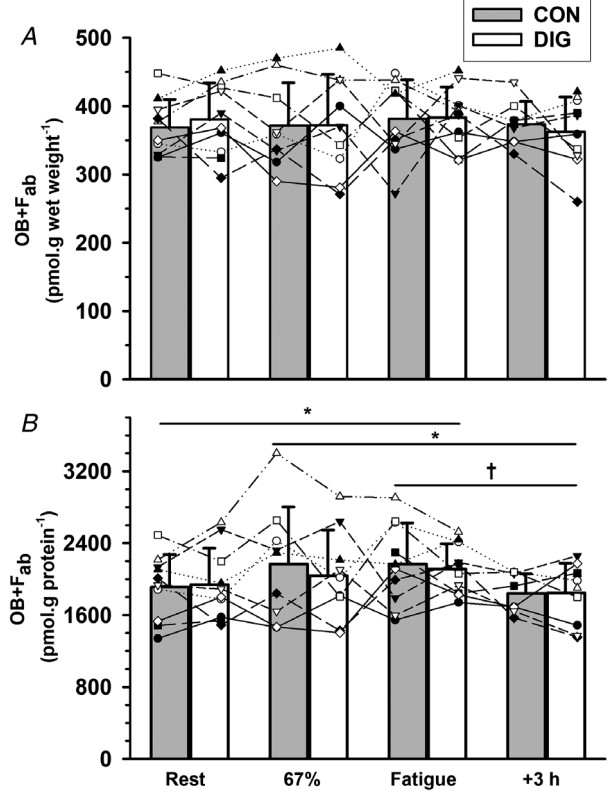

**Figure 3. Effects of digoxin and acute exercise on [³H]-ouabain binding site content in human skeletal muscle, measured after clearance of bound digoxin by $F_{ab}$ (OB+$F_{ab}$)**
The [³H]-ouabain binding site content measured after prior incubation in Digibind $F_{ab}$ (OB+$F_{ab}$), after 14 days of oral digoxin (DIG) or placebo (CON), and expressed relative to (*A*) muscle wet weight and (*B*) muscle protein content. Biopsies were taken at rest (Rest), after cycling for 10 min at 33% $V_{O_2peak}$ and 10 min at 67% $V_{O_2peak}$ (67%), immediately following cycling to fatigue at 90% $V_{O_2peak}$ (Fatigue) and at 3 h after exercise (+3 h). Data are mean (SD), *n* = 10; the exact number of data measurement points was 76. Individual data are plotted using different symbols and connected by dashed lines. Time effects are indicated as differences between the times connected by horizontal bar, *$P < 0.05$, †$P < 0.01$; a trend to higher OB+$F_{ab}$ (per g protein) at 67% *vs.* rest was found ($P = 0.087$, 21%). No significant digoxin effects ($P = 0.809$, $P = 0.376$), or time × treatment interaction ($P = 0.591$, $P = 0.861$) were found for OB+$F_{ab}$ expressed per g wet weight, or per g protein, respectively.

declined at fatigue (−15%, $P = 0.003$), before then increasing (*vs*. fatigue: $P = 0.016$) to return to rest levels by 3 h after exercise (Fig. 4*A*). No changes were found when 3-*O*-MFPase activity was expressed relative to muscle protein content (Fig. 4*B*). The calculated ratio of 3-*O*-MFPase activity/ouabain binding content was reduced from rest at both 67% V$_{O_2peak}$ ($P = 0.008$) and fatigue ($P = 0.003$) and recovered at 3 h after exercise.

*Digoxin.* The 3-*O*-MFPase activity did not differ between CON and DIG whether expressed relative to muscle wet weight ($P = 0.287$) or muscle protein ($P = 0.180$, Fig. 4). The ratio of 3-*O*-MFPase

activity/ouabain binding content did not differ between DIG and CON ($P = 0.770$).

## Quadriceps muscle strength and fatiguability

### Muscle strength

*Exercise.* Peak torque at each velocity during Cybex isokinetic contractions was highly reproducible during repeat variability trials (CV 4.3−7.9%). During torque–velocity testing (0–360° s$^{-1}$) in experimental

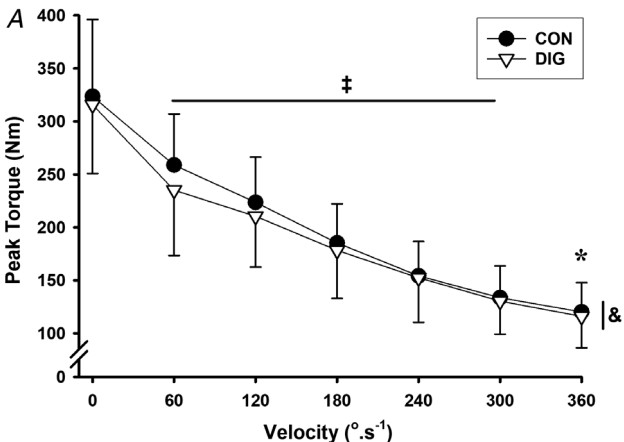

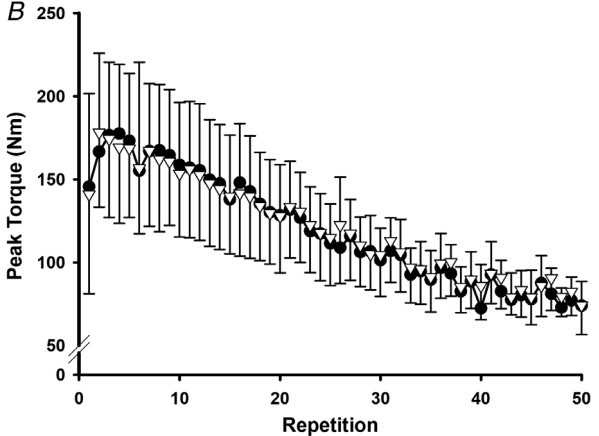

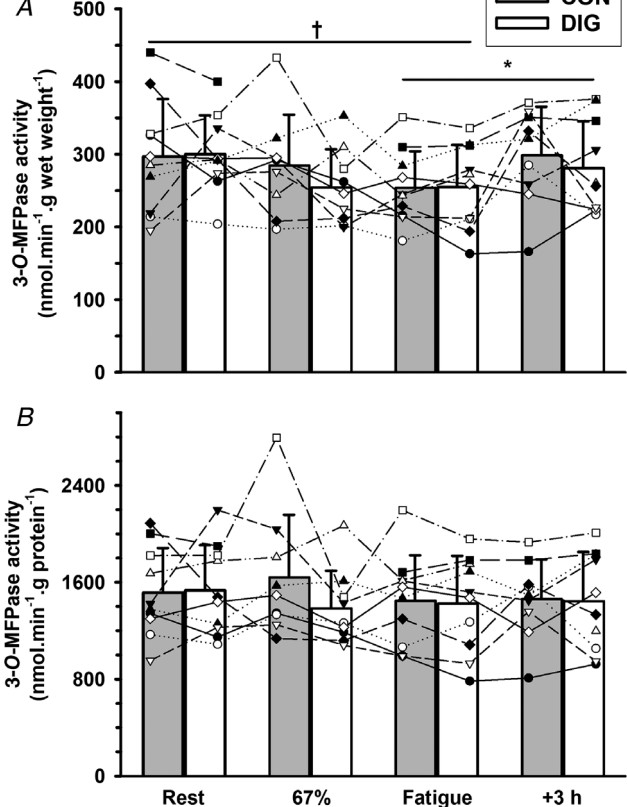

**Figure 4. Effects of digoxin and acute exercise on maximal *in vitro* 3-*O*-MFPase activity**

The maximal *in vitro* K$^+$-stimulated 3-*O*-MFPase activity after 14 days of digoxin (DIG) or placebo (CON) and expressed relative to (*A*) muscle wet weight and (*B*) muscle protein content. Biopsies were taken at rest (Rest), after cycling for 10 min at 33% V$_{O_2peak}$ and 10 min at 67% V$_{O_2peak}$ (67%), immediately following cycling to fatigue at 90% V$_{O_2peak}$ (Fatigue) and at 3 h after exercise (+3 h). Data are mean (SD), $n = 10$; the exact number of data measurement points was 77. Individual data are plotted using different symbols and connected by dashed lines. Time effects are indicated as differences between the times connected by horizontal bar, *$P < 0.05$, †$P < 0.01$, with exact $P$ values indicated in the Results. No significant digoxin effects ($P = 0.287$, $P = 0.180$), or time × treatment interaction ($P = 0.610$, $P = 0.321$) were found expressed per g wet weight, or per g protein, respectively.

**Figure 5. Effects of digoxin on quadriceps muscle strength and fatigue**

Quadriceps muscle peak torque–velocity from 0 to 360° s$^{-1}$ (*A*) and peak torque in each contraction during 50 isokinetic contractions repeated at 180° s$^{-1}$ (*B*) after digoxin (DIG, ∇) or placebo (CON, ●). Values are mean (SD), $n = 10$, except for the torque–velocity relationship at 180° s$^{-1}$, where $n = 9$; the exact number of data measurement points was 138 for torque–velocity and 1000 for fatigue data. For the torque–velocity relationship, time effects indicate the difference in torque from preceding velocity, *$P = 0.026$, ‡$P = 0.001$; treatment effect was DIG < CON (&$P = 0.010$). For the fatigue test, statistical differences are not indicated for simplicity, but peak torque during repetition #3 was greater than for repetition #6 ($P = 0.001$), repetition #9 ($P = 0.024$) and all subsequent repetitions ($P = 0.001$), with no difference between digoxin and placebo trials ($P = 0.221$).

**Table 1. Performance characteristics during finger flexion and leg cycling exercise trials**

| | Variability trials | | | Experimental trials | | |
|---|---|---|---|---|---|---|
| | 1 | 2 | CV (%) | DIG | CON | *P* |
| **Finger flexion** | | | | | | |
| Mean force (N) | 35.98 (10.51) | 36.04 (10.30) | 3.5 (2.3) | 37.37 (10.90) | 36.24 (11.07) | 0.438 |
| Mean power output (W) | 5.33 (1.07) | 5.29 (1.09) | 1.0 (1.0) | 5.44 (1.49) | 5.37 (1.52) | 0.729 |
| Time to fatigue (min) | 4.85 (2.65) | 4.90 (2.67) | 12.5 (10.6) | 2.62 (1.97) | 3.94 (3.37) | 0.197 |
| **Leg cycling** | | | | | | |
| Time to fatigue (min) | 5.69 (2.34) | 5.64 (2.27) | 5.9 (5.8) | 4.37 (2.59) | 4.23 (2.08) | 0.775 |

Two trials were conducted to determine variability followed by two experimental trials after 14 days of digoxin (DIG, 0.25 mg) or placebo (CON). Finger flexion exercise comprised three, 1 min bouts separated by 1 min of rest and a final bout continued to fatigue, at a power output corresponding to 105% $WR_{peak}$. Leg cycling exercise comprised 10 min at both 33% $\dot{V}_{O_2peak}$ and 67% $\dot{V}_{O_2peak}$ separated by 2 min of rest, then the final bout continued to fatigue at work rate corresponding to 90% $\dot{V}_{O_2peak}$. Values are mean (SD), *n* = 10 for power and force, *n* = 9 for time to fatigue during experimental trials. The coefficient of variation (CV) was calculated from individual performance during variability trials. *P* values for experimental trials are from a student *t* test.

trials, peak torque declined with each increased velocity up to 300° s$^{-1}$ ($P = 0.001$) and further at 360° s$^{-1}$ ($P = 0.026$).

*Digoxin.* Peak torque across all velocities was lower in DIG than in CON ($-4.3\%$, $P = 0.010$, Fig. 5*A*).

### Muscle fatigue

*Exercise.* The calculated fatigue index was highly reproducible during variability trials (CV = 4.7%). During experimental trials, the contraction peak torque declined markedly during the 50 repeated contractions at 180° s$^{-1}$ ($P = 0.001$, Fig. 5*B*).

*Digoxin.* There were no differences between trials in contraction peak torque during the 50 repetitions ($P = 0.221$), or in the calculated fatigue index [DIG, 53.6 (9.0) *vs.* CON, 57.4 (10.0)%, $P = 0.138$] (Fig. 5*B*).

### Finger flexion exercise results

#### FF performance

*FF peak incremental exercise.* During finger flexion incremental tests, the average power output at each incremental work rate was linear over time ($R^2 = 0.986$) and the incremental exercise $WR_{peak}$ was 5.10 (1.35) W.

*FF variability and experimental trials*

*Exercise.* During FF variability trials at 105% $WR_{peak}$, mean power output and force were highly reproducible (CV, 1.0% and 3.5%, respectively), whilst time to fatigue was more variable (CV, 12.5%).

*Digoxin.* None of these variables, including time to fatigue, differed between DIG and CON (Table 1).

#### FF plasma [K$^+$]

*Exercise.* Plasma [K$^+$]$_a$ increased above rest at PreEB2 ($P = 0.028$), PreEB3 ($P = 0.008$) and through until 2 min recovery ($P = 0.001$, Fig. 6*A*). Plasma [K$^+$]$_v$ rose sharply with each EB ($P = 0.001$) and decreased rapidly to rest values during each subsequent rest period; after exercise [K$^+$]$_v$ declined rapidly to below rest at 2 min ($P = 0.021$), 5 min ($P = 0.022$) and 10 min recovery ($P = 0.032$, Fig. 6*B*). Plasma [K$^+$]$_{a-v}$ decreased (i.e. became more negative) at EB1 ($P = 0.001$), representing net K$^+$ entry into plasma traversing forearm muscle, followed by an immediate reversal to near-rest values by PreEB2; [K$^+$]$_{a-v}$ subsequently followed this oscillating trend of net K$^+$ loss to plasma during ($P = 0.001$) EB2, EB3, EB4+1 and fatigue and reversal during intervening recovery periods; [K$^+$]$_{a-v}$ was more positive than rest at PreEB4 ($P = 0.006$), from 1 to 5 min ($P = 0.001$) and at 10 min recovery ($P = 0.004$, Fig. 6*C*). After correction for the corresponding $\Delta PV_{a-v}$, [K$^+$]$_{a-v}$ (corrected) displayed a similar oscillating pattern during each EB [$P = 0.001$; fatigue $-0.70$ (0.38) mmol l$^{-1}$], then immediate reversal to near-rest values at PreEB2 and PreEB3 and to values greater than rest (positive [K$^+$]$_{a-v}$), indicating a net K$^+$ loss from plasma at each of PreEB4 ($P = 0.031$), 1 min and 2 min ($P = 0.001$) and tendency at 5 min recovery ($P = 0.057$).

The K$^+$ fluxes across the forearm, calculated from plasma [K$^+$]$_{a-v}$(corrected) × plasma flow, indicated increased K$^+$ efflux into plasma during each EB ($P = 0.001$, fatigue, ~113-fold increase), with immediate reversals during intervening rest periods and recovery, indicating net K$^+$ removal from plasma (Table 2).

*Digoxin.* There were no effects of DIG on any of [K$^+$]$_a$ ($P = 0.524$), [K$^+$]$_v$ ($P = 0.147$), [K$^+$]$_{a-v}$ ($P = 0.477$)

**Table 2. Net K$^+$ fluxes into or out of plasma across the forearm musculature measured at rest, during intermittent finger flexion exercise continued to fatigue, and for 30 min of recovery, after 14 days of placebo (CON) and digoxin (DIG)**

| | Rest | Exercise | | | | | | | Recovery | | | | |
|---|---|---|---|---|---|---|---|---|---|---|---|---|---|
| | | EB1 | PreEB2 | EB2 | PreEB3 | EB3 | Pre EB4 | Fatigue | +1 | +2 | +5 | +10 | +30 |
| **CON** | | | | | | | | | | | | | |
| K$^+$$_{flux}$ | −1.2 | −95.4 | −0.6 | −106.2 | 4.2 | −104.5 | 7.5 | −125.1 | 19.0 | 14.7 | 1.7 | 0.7 | −0.7 |
| | (2.3) | (46.7)$^‡$ | (5.6) | (65.9)$^‡$ | (4.2) | (67.1)$^‡$ | (5.7) | (92.6)$^‡$ | (7.1) | (10.8) | (10.0) | (5.5) | (3.1) |
| **DIG** | | | | | | | | | | | | | |
| K$^+$$_{flux}$ | −0.6 | −104.0 | −1.6 | −97.7 | 8.2 | −100.1 | 5.8 | −112.2 | 14.8 | 8.3 | 0.7 | −1.8 | 0.4 |
| | (2.0) | (49.3)$^‡$ | (7.7) | (54.7)$^‡$ | (14.2) | (42.0)$^‡$ | (5.6) | (30.7)$^‡$ | (11.6) | (6.6) | (5.2) | (5.0) | (1.7) |

Fluxes calculated from ([ion]$_{a-v\ difference}$) × forearm plasma flow; K$^+$ fluxes were corrected for the arterio-venous $\Delta$PV. Units are $\mu$mol min$^{-1}$ for K$^+$ fluxes. A negative value denotes net ion influx into plasma across forearm musculature, and positive values denote net efflux from plasma across the musculature. Data are mean (SD), $n = 7$ for K$^+$ flux; the exact number of data measurement points was 168. Time effects indicate where different from rest. $^‡P = 0.001$. Treatment (digoxin) effect: no effects of digoxin were found for K$^+$ flux ($P = 0.865$).

(Fig. 6), [K$^+$]$_{a-v}$ (corrected) ($P = 0.359$) or K$^+$ efflux ($P = 0.865$) (Table 2). No differences were found between trials for the rise in plasma [K$^+$]$_a$ from rest to fatigue [$\Delta$[K$^+$]$_a$, DIG, 0.34 (0.36); CON, 0.36 (0.22) mmol l$^{-1}$, $P = 0.896$], or the post-exercise decline in [K$^+$]$_a$ from fatigue during 5 min recovery [$-\Delta$[K$^+$]$_a$, DIG, −0.30 (0.25); *vs.* CON, −0.27 (0.25) mmol l$^{-1}$, $P = 0.816$].

*FF forearm blood flow, muscle CO$_2$ output and muscle O$_2$ uptake*

*Exercise.* Forearm blood flow was increased throughout the exercise period and until 5 min ($P = 0.001$) and 10 min recovery ($P = 0.025$); flow rose ∼12-fold from rest to fatigue and declined by ∼60% from fatigue within the first minute of recovery ($P = 0.001$). The calculated plasma flow was similarly increased throughout the exercise period until 5 min of recovery ($P = 0.001$) and tended to be elevated at 10 min ($P = 0.055$). The forearm muscle CO$_2$ output ($V_mCO_2$) increased during EB1 ($P = 0.001$), PreEB2 ($P = 0.015$), EB2 ($P = 0.001$), PreEB3 ($P = 0.014$) and until fatigue ($P = 0.001$, ∼22-fold increase). The forearm muscle O$_2$ uptake ($V_mO_2$) increased during EB1 ($P = 0.002$), PreEB2 ($P = 0.05$) and from EB2 until fatigue ($P = 0.001$, ∼24-fold increase).

*Digoxin.* Forearm blood flow (−12.7 ml min$^{-1}$, treatment effect, $P = 0.019$) and calculated plasma blood flow (−3.8 ml min$^{-1}$, $P = 0.020$) were lower in DIG than in CON. There was a tendency for reduced $V_mCO_2$ with DIG (−3.9 ml min$^{-1}$, $P = 0.078$), but no effect of DIG on $V_mO_2$ ($P = 0.202$).

*FF [Hb], Hct, $\Delta$PV and $\Delta$BV*

*Exercise.* [Hb]$_a$ was elevated above rest at PreEB4 ($P = 0.045$), fatigue ($P = 0.002$), 1 min ($P = 0.004$),

2 min ($P = 0.008$) and 5 min recovery ($P = 0.041$); [Hb]$_v$ was increased during EB1 and EB2 ($P = 0.006$), EB3 ($P = 0.007$), EB4+1 ($P = 0.003$), at fatigue ($P = 0.01$) and at 1 min recovery ($P = 0.036$). Hct$_a$ increased at EB3 ($P = 0.032$), PreEB4 ($P = 0.016$), EB4+1 ($P = 0.004$), fatigue through to 2 min ($P = 0.001$) and 5 min recovery ($P = 0.005$); Hct$_v$ was increased during EB1 ($P = 0.001$), EB2 ($P = 0.005$), EB3, EB4+1 and fatigue ($P = 0.001$), at 1 min ($P = 0.005$) and at 2 min recovery ($P = 0.034$). PV$_a$ declined from rest ($\Delta$PV$_a$) during EB1 ($P = 0.029$), PreEB2 ($P = 0.019$), EB2 ($P = 0.001$), PreEB3 ($P = 0.002$) and EB3 through to 5 min ($P = 0.001$) and at 10 min recovery ($P = 0.023$). Similarly, $\Delta$PV$_v$ fell throughout the exercise period and until 2 min ($P = 0.001$) and at 5 min recovery ($P = 0.035$). A negative $\Delta$PV$_{a-v}$ indicated a small net loss in PV across the forearm during each of the exercise bouts ($P = 0.001$). BV$_a$ declined below rest at EB2 ($P = 0.002$), PreEB3 ($P = 0.004$) and from EB3 (fatigue, −2.7 (1.9)%) through until 5 min recovery ($P = 0.001$). Similarly, $\Delta$BV$_v$ declined below rest at EB1 through until 2 min recovery ($P = 0.001$). A negative $\Delta$BV$_{a-v}$ indicated a small net loss in BV across the forearm; $\Delta$BV$_{a-v}$ declined below rest during each EB ($P = 0.001$) and PreEB2 ($P = 0.017$).

*Digoxin.* [Hb]$_a$ was slightly higher in DIG than in CON (0.1 g dl$^{-1}$, $P = 0.026$), with no differences found between trials for [Hb]$_v$ ($P = 0.506$), Hct$_a$ ($P = 0.463$) or Hct$_v$ ($P = 0.992$). During DIG, $\Delta$PV$_a$ was more negative (−1.4%, $P = 0.001$) and $\Delta$PV$_v$ tended to be less negative (0.6%, $P = 0.054$) than in CON, whilst $\Delta$PV$_{a-v}$ did not differ between trials ($P = 0.167$). During DIG, $\Delta$BV$_a$ was more negative (−0.8%, $P = 0.001$) whilst $\Delta$BV$_v$ was less negative (0.6%, $P = 0.007$) than in CON; $\Delta$BV$_{a-v}$ did not differ between trials ($P = 0.198$).

## Leg cycling test results

### LC exercise performance, $V_{O_2peak}$ and heart rate

*LC incremental exercise test.* The LC incremental exercise $V_{O_2peak}$ and $WR_{peak}$ were 3.67 (0.42) l min$^{-1}$ and 298 (23) W, respectively.

*LC variability and experimental trials.* During LC variability trials, good reproducibility was seen for $VO_2$ during each of 33% (CV, 2.2%), 67% (CV, 1.9%) and 90% $V_{O_2peak}$ (CV, 4.4%), as well as for the time to fatigue at 90% $V_{O_2peak}$ (CV, 5.9%). During LC experimental trials, no significant differences were found between DIG and

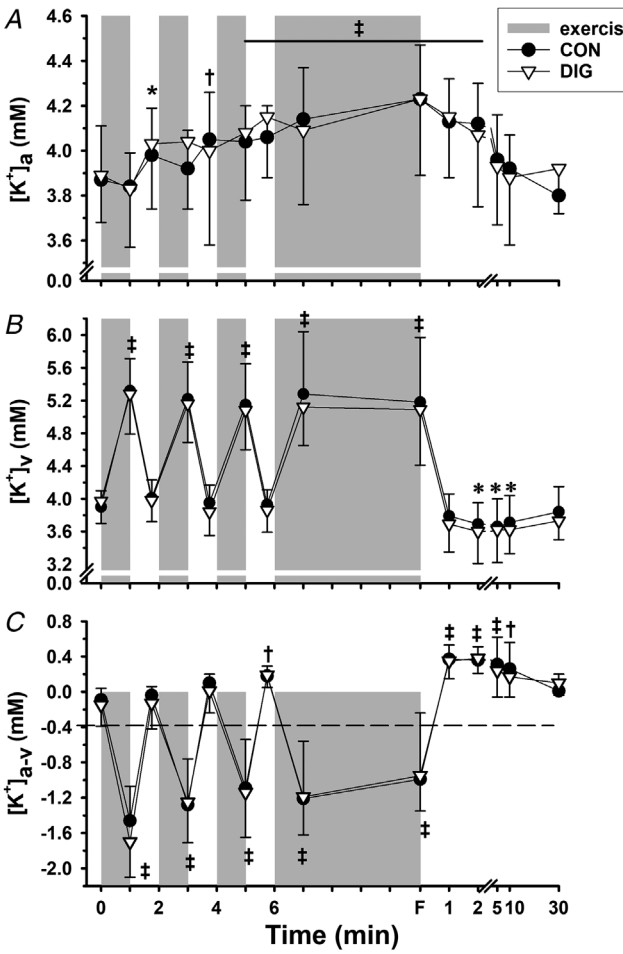

**Figure 6. Effects of digoxin on K$^+$ regulation during and after finger flexion exercise**

Plasma [K$^+$] at rest, during and following intense, intermittent finger flexion exercise after digoxin (DIG, $\nabla$) or placebo (CON, $\bullet$). Panels indicate arterial (*A*), venous (*B*) and calculated arteriovenous difference (a-v) uncorrected (*C*) for plasma [K$^+$]. Exercise is denoted by shaded bars; intermittent exercise and intervening rest time, F fatigue, and recovery time after fatigue. Data are expressed as mean (SD); sample size is as indicated in the Methods. Time effects indicate where different from rest, *$P < 0.05$; †$P < 0.01$; ‡$P < 0.001$, with exact *P* values indicated in the Results. A treatment (DIG) effect and time × treatment interaction were not significant.

CON for the time to fatigue at 90% $V_{O_2peak}$ ($P = 0.775$, Table 1), for VO$_2$ [33% $V_{O_2peak}$, DIG, 1.21 (0.20) *vs.* CON, 1.19 (0.21); 67% $V_{O_2peak}$, DIG, 2.63 (0.51) *vs.* CON, 2.68 (0.54); 90% $V_{O_2peak}$, DIG, 3.56 (0.28) *vs.* CON, 3.55 (0.67) l min$^{-1}$, respectively], or heart rate [rest, DIG, 80 (4) *vs.* CON, 77 (5); fatigue, DIG, 193 (5) *vs.* CON, 191 (6); 30 min recovery DIG, 94 (7) *vs.* CON, 93 (5) bpm].

### LC plasma [K$^+$]

*Exercise.* Plasma [K$^+$]$_a$ increased above rest at EB2+1, EB2, EB3+1 and at fatigue where it peaked ($P = 0.001$), but did not differ from rest during the intervening rest periods; [K$^+$]$_a$ decreased rapidly after fatigue but remained elevated at 1 min ($P = 0.001$), fell below rest at 5 min ($P = 0.012$) and tended to be less also at 10 min recovery ($P = 0.082$; Fig. 7A). Plasma [K$^+$]$_v$ increased above rest from EB2 through fatigue until 2 min recovery (all $P = 0.001$, except PreEB3, $P = 0.002$, Fig. 7B). Plasma [K$^+$]$_{a-v}$ was mostly positive during the exercise period, indicating net K$^+$ uptake into inactive forearm muscle, but was greater than rest only at fatigue ($P = 0.001$), with a tendency to also increase during EB3+1 ($P = 0.059$, Fig. 7C). After correction for the corresponding $\Delta PV_{a-v}$, [K$^+$]$_{a-v}$ (corrected) displayed a similar pattern, tending to be greater than rest at EB2+1 ($P = 0.058$), EB2 ($P = 0.053$), EB3+1 ($P = 0.053$), increased at fatigue [1.71 (1.11), $P = 0.001$] and at 2 min recovery ($P = 0.008$) and tending to be less at 5 min recovery ($P = 0.073$).

*Digoxin.* There was no effect of DIG on plasma [K$^+$]$_a$ ($P = 0.833$, Fig. 7A), whereas [K$^+$]$_v$ was lower in DIG than in CON (0.15 mmol l$^{-1}$, $P = 0.042$, Fig. 7B), with a greater (more positive) [K$^+$]$_{a-v}$ in DIG compared to CON (0.08 mmol l$^{-1}$, $P = 0.004$, Fig. 7C); [K$^+$]$_{a-v}$ (corrected) did not differ between trials ($P = 0.377$). No differences were found between trials for the rise in [K$^+$]$_a$ from rest to fatigue [$\Delta$[K$^+$]$_a$, DIG, 2.59 (0.60) *vs.* CON, 2.65 (0.74), $P = 0.675$], or the decline from fatigue to 5 min recovery [$-\Delta$[K$^+$]$_a$, DIG, $-2.88$ (0.65) *vs.* CON, $-2.95$ (0.64), $P = 0.667$].

### LC haematology and fluid shifts

*LC Hb, Hct, $\Delta PV$ and $\Delta BV$*

*Exercise.* [Hb]$_a$ increased above rest at EB2+1 ($P = 0.003$) and through until 10 min recovery ($P = 0.001$), whilst [Hb]$_v$ was increased at PreEB2 ($P = 0.018$) and from EB2 until 10 min recovery ($P = 0.001$). Hct$_a$ was increased above rest from EB2+1 ($P = 0.002$) and then until 10 min recovery ($P = 0.001$), whilst Hct$_v$ was increased at PreEB2, EB2+1 ($P = 0.006$) and from EB2 until 10 min recovery ($P = 0.001$). The arterial plasma volume declined from rest ($\Delta PV_a$) during EB1+1 ($P = 0.007$), EB1 ($P = 0.008$), from EB2+1 through to fatigue and until 10 min recovery ($P = 0.001$);

$\Delta PV_a$ was negative until 30 min recovery, where it was elevated above rest [2.3 (2.8)%, $P = 0.016$]. $\Delta PV_v$ similarly declined during EB1+1 ($P = 0.023$), PreEB2 ($P = 0.001$), EB2+1 ($P = 0.004$), through to fatigue and until 10 min recovery ($P = 0.001$). $\Delta PV_{a-v}$ across the forearm was positive during exercise bouts, indicating a small net gain in fluid, but these did not differ from rest; negative values were seen at PreEB2 ($P = 0.025$), PreEB3 ($P = 0.005$), 10 min ($P = 0.009$) and 30 min recovery ($P = 0.008$).

Similar exercise effects were seen for $\Delta BV_a$, $\Delta BV_v$ and $\Delta BV_{a-v}$ as for the corresponding $\Delta PV$, $\Delta PV_v$ and $\Delta PV_{a-v}$.

*Digoxin.* There were no differences between trials for arterial or venous [Hb] or Hct. $\Delta PV_a$ declined more in DIG than in CON ($\sim$1% more negative, $P = 0.022$), whilst there were no effects of DIG on $\Delta PV_v$ ($P = 0.869$) or $\Delta PV_{a-v}$ ($P = 0.669$). $\Delta BV_a$ declined more in DIG than in CON ($\sim-0.5$% more negative, $P = 0.048$); $\Delta BV_v$ tended to decline more in DIG ($\sim-0.5$%, $P = 0.083$), with no effects of DIG on $\Delta BV_{a-v}$ ($P = 0.699$).

## Discussion

This study investigated the effects of a typical clinical oral dose of the cardiotonic steroid digoxin taken for 14 days by healthy adults, on muscle NKA and on plasma [K$^+$] during and after intensive, two-legged cycling and finger flexion exercise continued to fatigue, and on muscle strength, with six novel findings. First, even though serum [digoxin] rose to within the clinical therapeutic range, muscle NKA was preserved with digoxin, with unchanged muscle [$^3$H]-ouabain binding site content (OB-F$_{ab}$) and maximal *in vitro* 3-*O*-MFPase activity after digoxin. Second, OB-F$_{ab}$, which is normally a measure of total NKA content in human muscle, was further increased after digoxin (8.2%), when measured after pre-incubation of muscle in Digibind digoxin antibody fragments (OB+F$_{ab}$), with a calculated 7.6% digoxin–NKA occupancy in muscle. Hence, in healthy participants, the expected muscle NKA deficit due to digoxin binding of NKA in muscle did not occur, in contrast to findings in cardiac patients. This resilience of NKA after digitalisation points to the preservation of muscle NKA as an important skeletal muscle strategy upon challenge. Third, muscle OB-F$_{ab}$ increased with acute exercise at 67% V$_{O_2peak}$ when expressed both per wet weight and per protein and also at fatigue when expressed per protein. These changes, which occurred in a 20–30 min timeframe, might reflect rapid assembly of existing subunit isoforms; an increase in OB+F$_{ab}$ (per protein) was found at fatigue. Fourth, the quadriceps muscle strength was reduced with digoxin, which has important implications for clinical use of digoxin. Fifth, despite elevated plasma [digoxin], arterial [K$^+$] during exhaustive dynamic exercise was not reduced and K$^+$ homeostasis during and following exercise was not impaired, which contrasts with changes previously reported in cardiac patients. Of the K$^+$ variables measured, only the [K$^+$] arterio-venous difference across inactive forearm muscle was altered with digoxin, being slightly increased during LC exercise. Sixth, consistent with the preservation of muscle NKA content and minimal effects of digoxin on [K$^+$] despite the elevated [digoxin], fatiguability was also unchanged with digoxin,

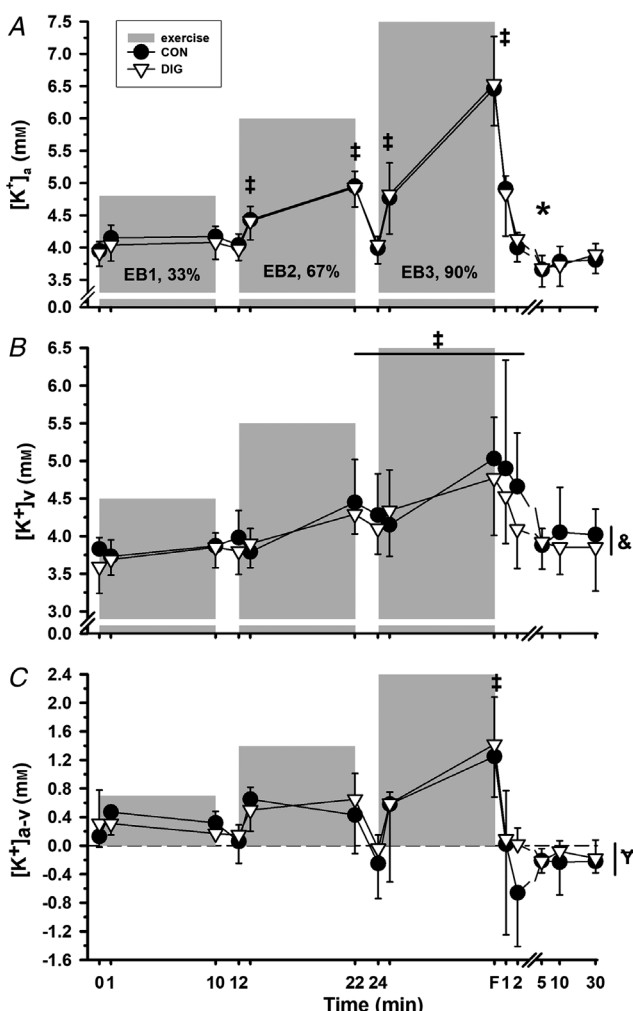

**Figure 7. Effects of digoxin on plasma K$^+$ regulation during and after leg cycling exercise**

Plasma [K$^+$] at rest, during and following two-legged cycling exercise after digoxin (DIG, $\nabla$) or placebo (CON, $\bullet$). Panels indicate arterial (*A*), venous (*B*) and calculated arteriovenous difference (a-v) uncorrected (*C*) for plasma [K$^+$]. Exercise is denoted by shaded bars; discontinuous exercise and intervening rest time, F fatigue, and recovery time after fatigue. Data are expressed as mean (SD); sample size is as indicated in the Methods. Time effects indicate where different from rest, *$P < 0.05$; †$P < 0.01$; ‡$P < 0.001$, with exact $P$ values indicated in the Results. A treatment (DIG) effect indicated as different to CON, where $^\&P = 0.042$, $^¥P = 0.004$. The time × treatment interaction was not significant.

as measured using three different intense exercise models. Thus, despite clinically relevant elevation of digoxin, there was strong preservation of muscle NKA, $K^+$ regulation and exercise performance, also with corresponding minor effects on metabolism and acid–base balance with exercise.

### Clinically relevant serum digoxin concentration

Oral digoxin administration of 0.25 mg $day^{-1}$ for 14 days raised serum [digoxin] to 0.8 nmol $l^{-1}$ ($\sim$0.62 ng $ml^{-1}$), with no difference between days 7 and 14, similar to a previous report using the same dose in healthy males (Schenck-Gustafsson et al., 1987) and only $\sim$20% less than was found after taking twice this digoxin dosage (0.5 mg $day^{-1}$) for 14 days in healthy individuals (Sundqvist et al., 1983). Importantly, the serum [digoxin] attained was within the optimal therapeutic range of $\sim$0.65–1.15 nmol $l^{-1}$ ($\sim$0.5–0.9 ng $ml^{-1}$) (Bavendiek et al., 2017). Thus, our digoxin protocol achieved the desired outcome of a typical clinical [digoxin], allowing us to then investigate the consequent physiological and functional impacts on muscle NKA and plasma $K^+$ regulation in healthy participants.

### Preservation of NKA content in skeletal muscle despite elevated digoxin

These are the first measures of digoxin effects on muscle $[^3H]$-ouabain binding site content (OB-$F_{ab}$) in healthy individuals, and the first serial measures of OB-$F_{ab}$ taken both before and following digoxin treatment, with a major finding being that OB-$F_{ab}$ did not decline after 14 days of digoxin treatment in healthy adults. The typical OB-$F_{ab}$ assay measures $[^3H]$-ouabain binding to all functional $\alpha\beta$ complexes, which, in human skeletal muscle, normally enables complete quantification of NKA and is therefore the gold standard measure of NKA content (McKenna et al., 1993; Nørgaard et al., 1983). We hypothesised that digoxin binding to, and inhibition of, NKA in skeletal muscle (Jogestrand & Sundqvist, 1981; Joreteg, 1986) would reduce the OB-$F_{ab}$ in resting muscle, based on reports of substantially reduced OB-$F_{ab}$ in muscle in digitalised heart failure patients (Schmidt et al., 1991, 1995; Schmidt, Allen et al., 1993). Also, another study on heart failure patients receiving digoxin reported similar muscle OB to controls, but these analyses were conducted after first removing bound digoxin with Digibind, thus indicating that pre-Digibind OB (i.e. OB-$F_{ab}$) must have been substantially reduced with digitalisation in their cohort (Green et al., 2001). Our findings of unchanged muscle OB-$F_{ab}$ in healthy participants with digoxin therefore clearly contrast earlier studies reporting reduced muscle OB-$F_{ab}$ in heart failure patients with digitalisation.

In muscle exposed to digoxin, a fraction of NKA already bound by digoxin is unavailable for $[^3H]$-ouabain binding, thereby underestimating true NKA content (Schmidt & Kjeldsen, 1991). Hence, we also determined the digoxin–NKA occupancy using digoxin antibody fragments ($F_{ab}$) to first clear any digoxin bound in muscle, followed by the standard ouabain binding measures (OB+$F_{ab}$) (Schmidt & Kjeldsen, 1991). We confirmed that this $F_{ab}$ wash methodology enabled almost complete NKA recovery (96%) in muscle biopsy samples from healthy humans incubated in digoxin, almost identical to that reported in skeletal muscle obtained post-mortem (Schmidt & Kjeldsen, 1991) and in heart failure patients (Green et al., 2001). A key finding here was that when measured after Digibind incubation, the ouabain binding sites in resting muscle samples had in fact increased by 8.2% after 14 days of digoxin (i.e. OB+$F_{ab}$ was 8.2% greater than OB-$F_{ab}$ in DIG). This indicated a greater overall total NKA (i.e. OB+$F_{ab}$) existed in muscle after digitalisation in these healthy participants, which was not the case in cardiac patients. The muscle NKA–digoxin occupancy in these healthy participants was 7.6% of the total ouabain binding sites (i.e. 7.6% of OB+$F_{ab}$). This is consistent with the 9% occupancy reported in muscle from heart failure patients digitalised for 3 days, and although lower, also with the 13% occupancy in muscle obtained post-mortem from previously digitalised patients (Schmidt et al., 1995; Schmidt, Holm-Nielsen et al., 1993). The higher occupancy in these two clinical studies is probably due to the patient's higher serum [digoxin] of 1.2–2.3 nм (Schmidt et al., 1995; Schmidt, Holm-Nielsen et al., 1993), already lower muscle NKA content with heart failure (Norgaard et al., 1990), and probably reduced muscle mass due to sarcopenia and/or cachexia. Importantly, none of these studies contrasted digoxin occupancy before *vs.* after digitalisation within the same individuals. Our findings may therefore also give some insights into understanding muscle adaptation in heart failure patients. It is possible that NKA synthesis evident in healthy muscle is diminished in advanced heart failure, eventually contributing to reduced muscle NKA content (Norgaard et al., 1990). This would also be consistent with their lack of muscle NKA upregulation in response to physical training, as NKA content actually declined after 3–4 months of training (Green et al., 2001), in contrast to the consistent NKA upregulation with training in healthy adults (Wyckelsma et al., 2019). Our findings of preservation of NKA content and maximal 3-*O*-MFPase activity with digoxin in resting muscle are also internally consistent. Since 3-*O*-MFPase activity was not also measured with Digibind, it is not possible to determine whether this might also have been greater after digoxin removal. Despite the uncertainties with maximal 3-*O*-MFPase activity as a measure of NKA activity after exercise (Juel et al., 2013), we have utilised

this measure here as additional evidence verifying the lack of depression in NKA after digoxin treatment. For future studies, alternative measures of NKA activity would be beneficial to further explore the functional effects of NKA inhibition. In rats, high-dose digoxin initially depressed 3-$O$-MFPase activity in muscle by 13% after 3–7 days, but this then recovered after 3 months of further digoxin treatment, suggesting NKA adaptability (Li et al., 1993). Interestingly, the 8.2% gain in OB+F$_{ab}$ in human muscle after 14 days of digoxin is substantially less than the 30% elevation in NKA activity found in guinea pig myocardium after digitoxin for 10–15 days (Bonn & Greeff, 1978). Further research on the acute and chronic effects of digoxin on NKA in muscle is of interest not only because of digoxin's ongoing clinical usage, but also given the growing focus on cardiotonic steroid signal transducing, protein–protein interactions and intracellular signalling functions via Src-dependent and Src-independent pathways (Aperia et al., 2016; Cui & Xie, 2017).

There are several interpretations for why muscle OB-F$_{ab}$ was maintained and not depressed with 14 days of digoxin. The first is a possible compensatory upregulation of NKA in muscle after digoxin, which acted to preserve muscle functional NKA content and would be consistent with both the unchanged OB-F$_{ab}$ and the 8.2% greater OB+F$_{ab}$ after digoxin. This could be due to increased synthesis and/or reduced catabolism of NKA subunits, but several of our findings are inconsistent with this. These include first that OB+F$_{ab}$ in DIG was not greater than in CON as would be expected (Fig. 3; the relevant interaction effect was non-significant). Second, increases in protein abundances of the dominant NKA isoforms would be expected after digoxin, namely $\alpha_1$, $\alpha_2$, $\beta_1$ or $\beta_2$ isoforms, whereas in fact, no significant changes were detected (our unpublished data). Another finding inconsistent with possible upregulation of muscle NKA with digoxin is the lack of effect on muscle NKA $\alpha_{1-3}$ or $\beta_{1-3}$ isoform mRNA expression, with no effects evident after 14 days (our unpublished data). This was despite digoxin occupancy of 7.6%, and contrasts with other reports that cardiac glycosides differentially affect NKA isoform mRNA and transcription rates (Kometiani et al., 2000; Murphy et al., 2006; Wang et al., 2000; Yamamoto et al., 1993). However, because those studies used different species, tissues and cell preparations, with different glycoside concentrations and also with conflicting findings, it is difficult to compare their findings to ours in skeletal muscle in healthy humans. It is possible that an initial effect of NKA inhibition due to digoxin binding did affect individual NKA isoform gene transcripts, but this cannot be answered without a time-course study. Therefore, at this time we cannot satisfactorily explain how muscle OB was sustained in the face of elevated serum and presumably also tissue levels of digoxin. Further experiments are required to ascertain whether any NKA isoforms are upregulated at the protein level to correspond with the greater muscle OB after digoxin (OB+F$_{ab}$). It should also be noted, however, that no NKA upregulation is also inconsistent with the calculated digoxin binding in muscle of ∼29 pmol g$^{-1}$ wet weight. This digoxin binding was greater than muscle digoxin content measured by radioimmunoassay in earlier studies, of ∼8–10 pmol g$^{-1}$ wet weight (∼32 −39 nmol kg$^{-1}$ dry weight) in digitalised healthy males with a similar [digoxin] (Ericsson et al., 1981; Jogestrand & Sundqvist, 1981; Schenck-Gustafsson et al., 1987). In future studies, an initial muscle biopsy taken after an acute dose of digoxin would clarify the extent of initial digoxin binding and allow measures of OB+F$_{ab}$ without any possible compensatory upregulation.

It seems unlikely that post-translational modifications of NKA due to glutathionylation contributed to our findings of greater OB+F$_{ab}$ than OB-F$_{ab}$ in muscle. Studies in numerous cell types have shown that a fraction of NKA subunits are glutathionylated under basal conditions and that glutathionylation can be rapidly and reversibly increased with oxidative stress (Petrushanko et al., 2012, 2017), which inactivates NKA and therefore can regulate NKA activity (Dergousova et al., 2018). Glutathionylation of NKA $\alpha$ subunits and of $\beta_1$ and $\beta_2$ isoforms also occurs in human skeletal muscle, and increased glutathionylation via GSSG incubation was associated with decreased NKA activity (Juel et al., 2015). Hence a fraction of NKA in muscle may be inhibited by glutathionylation, with implications for functional NKA content and activity, although the regulation is complex (Pirkmajer & Chibalin, 2016), as glutathionylation of FXYD-1 may exert protective effects against glutathionylation of NKA sub-units (Bibert et al., 2011). Ouabain fixes NKA in the E2P conformation, thereby reducing cysteine availability for glutathionylation and protecting NKA from inactivation (Poluektov et al., 2019). High-dose ouabain (10 $\mu$M) reduced both glutathionylation and NKA activity in heart homogenates (Petrushanko et al., 2012), but the effects of nanomolar ouabain, or of digoxin, on NKA glutathionylation are unclear. However, it would seem likely that, if anything, digoxin binding in muscle may in fact reduce NKA glutathionylation, thereby removing NKA inhibition and increasing NKA availability. Hence, application of Digibind to remove digoxin would not then allow detection of previously inactive digoxin-bound NKA due to glutathionylation, but perhaps the opposite.

An important finding was that a significant elevation in OB+F$_{ab}$ was found only after digoxin, not in control muscle, where a non-significant difference with F$_{ab}$ of 4.5% was found, similar to earlier findings of 5–7% in muscle incubated in F$_{ab}$ but without digoxin exposure (Schmidt & Kjeldsen, 1991; Schmidt, Holm-Nielsen et al., 1993; Green et al., 2001). We suggest that this low 'occupancy' value of 4.3% in controls probably mainly

reflects variability in the OB method, but that it might also include some actual NKA occupancy, via binding to endogenous ouabain or ouabain-like compounds in muscle. In humans, plasma [ouabain] of between 0.05 and 1 nм and plasma [marinobufagenin] of between 0.2 and 0.6 nм have been reported (Orlov et al., 2020), with much higher values also reported for ouabain-like compounds of 2.5 and of 7.3 nм for an 'NKA-inhibitor' in plasma for humans at rest, both of which were markedly elevated with exercise (Bauer et al., 2005). Although Digibind does bind both endogenous ouabain and ouabain-like compounds (Pullen et al., 2004), we cannot determine whether this also occurred here, as we did not measure these compounds in plasma or muscle. However, if present, they would contribute to the low apparent occupancy in control muscle in these healthy participants, but also to the calculated digoxin occupancy in the DIG muscle. However, this possibility remains to be determined.

## Acute exercise effects on NKA content and 3-*O*-MFPase activity

A novel finding was the transient and rapid increase in muscle OB-$F_{ab}$ after only 20–30 min of exercise. The OB-$F_{ab}$ was elevated by 10% after 20 min of exercise comprising 10 min each at 33% and 67% $V_{O_2 peak}$, when expressed per gram wet weight (NS, 5% at fatigue) and by 21% when expressed per gram protein, as well as by 16% at fatigue when expressed per gram protein. These intriguing findings indicate that muscle NKA content was acutely increased with exercise. It is unlikely that this can be explained by any of increased NKA synthesis and/or reduced degradation, increased recruitment of NKA subunits to the sarcolemma and t-tubules from intracellular stores, increased rate of binding of ouabain due to elevated NKA activity, methodological artefacts and/or digoxin effects, as detailed below. We suggest alternatively that this might reflect increased formation of functional NKA complexes from already existing, but non-bound, $\alpha$ and $\beta$ subunits in muscle.

The elevated muscle OB-$F_{ab}$ with exercise differs from our previous reports that OB-$F_{ab}$ was unchanged with acute exercise (see references in McKenna et al., 2008), but appears consistent with the 13% increase reported in muscle OB-$F_{ab}$ after prolonged running for ~10 h and attributed to NKA synthesis (Overgaard et al., 2002). However, increased NKA synthesis or decreased degradation seems unlikely. The increase here occurred after only 20–30 min of exercise, where it is unlikely that muscle could synthesise ~35–70 pmol NKA (per gram muscle).

The rapid increase in OB-$F_{ab}$ with exercise is also unlikely to simply reflect an increased *rate* of ouabain binding in muscle due to increased NKA activity, due to neuro-humoral responses to exercise or elevated intracellular [$Na^+$] with muscle contractions. First, the assay utilises 2 h of incubation to ensure saturation of [$^3$H]-ouabain binding occurs (Nørgaard et al., 1984). Second, in rat muscle, saturation of [$^3$H]-ouabain binding and also unchanged OB-$F_{ab}$ were evident under conditions of elevated NKA activity due to electrical stimulation or insulin, where an increase in OB-$F_{ab}$ would be expected if NKA activity or availability of NKA had increased (McKenna et al., 2003). In that study, ouabain binding was greater in the early period of incubation, but this effect had disappeared with a plateau after 2 h of incubation. This suggests that a functional elevation in pump activity cannot account for our results of increased OB-$F_{ab}$ with exercise.

In non-digitalised human skeletal muscle biopsy pieces, the typical ouabain binding assay fully quantifies NKA (Nørgaard et al., 1984) and in rodent muscle, binding does not differ between intact muscles and cut muscle pieces (Clausen, 2013). Thus, OB-$F_{ab}$ should measure all functional NKA in pieces of human muscle (Clausen, 2013). The OB-$F_{ab}$ method was recently criticised as being too slow to detect increased NKA recruitment due to translocation, as these pumps would then already be included in the final OB-$F_{ab}$ measure (Pirkmajer & Chibalin, 2016). This possibility cannot explain our finding because OB-$F_{ab}$ was increased with exercise. Therefore, it seems unlikely that the increase in OB-$F_{ab}$ with exercise could be due to increased recruitment to the sarcolemma and t-tubules of otherwise unavailable NKA, or of NKA subunit isoforms (Benziane & Chibalin, 2008; Pirkmajer & Chibalin, 2016) from undefined intracellular stores, as reported in membrane fractions following exercise (Tsakiridis et al., 1996; Juel, Nielsen et al., 2000; Juel et al., 2001), or possibly from caveolae (Kristensen et al., 2008). The increase in OB-$F_{ab}$ with exercise also cannot simply be due to fluid shifts, since dilution might be expected to reduce OB-$F_{ab}$ per wet weight but not change OB-$F_{ab}$ per protein. We suggest that this also cannot simply be excluded as a methodological artefact, since an increase in ouabain binding with exercise was similarly detected for both OB-$F_{ab}$ and OB+$F_{ab}$ when expressed per gram protein. Our finding is also unexpected with digoxin, as we anticipated that OB-$F_{ab}$ might even decline with exercise in DIG, since digoxin binding to muscle was previously reported to increase by up to 20% during exercise (Joreteg & Jogestrand, 1983). In that study, however, the digoxin dosage taken was double that of the present study (i.e. 0.5 *vs.* 0.25 mg day$^{-1}$, for 14 days) and the serum [digoxin] achieved was 75–88% higher than in our study (i.e. 1.4–1.5 *vs.* 0.8 nм). Whether this explains the discrepancy between the two studies on acute exercise effects on muscle NKA content is unclear.

One possibility is that structural alterations to NKA with exercise, caused by glutathionylation or

phosphorylation of NKA subunits or of phospholemman might enable these increases in OB-F$_{ab}$ with exercise. Glutathionylation of NKA $\beta_1$, but not $\alpha$ subunits or $\beta_2$ isoforms, increased in muscle with intense exercise and was also associated with reduced NKA activity measured by both direct Pi release and the 3-$O$-MFPase method (Juel et al., 2015). However, whilst fractional NKA inactivation could reduce muscle OB-F$_{ab}$, this is opposite to the effect seen here with exercise. It therefore seems unlikely that NKA glutathionylation or de-glutathionylation is responsible for the increase found in OB-F$_{ab}$ with exercise. Whilst phosphorylation of phospholemman can increase NKA activity and also increases with muscle contraction (Pirkmajer & Chibalin, 2016), this also seems unlikely to account for the increased OB-F$_{ab}$ with exercise. Glycosylation of the $\beta$-subunit is essential for NKA functional expression in cultured chick muscle cells, as measured via ouabain binding and activity (Alboim et al., 1992), but it is unknown whether $\beta$-glycosylation is modified by muscle contraction and therefore whether this also affects NKA activity or ouabain binding.

We therefore consider an alternative possible mechanism for the elevated OB-F$_{ab}$ with acute exercise, in the absence of increases in $\alpha$ subunit proteins. We suggest that an increased formation of functional NKA complexes occurred from already existing, but non-bound, $\alpha$ and $\beta$ subunits in muscle, that were either inserted into, or being trafficked to, sarcolemmal/t-tubular membranes (DeTomaso et al., 1994) that could then be detected by ouabain binding. Further research using different techniques is required to test this possibility.

Our findings of acute exercise effects on muscle NKA were somewhat paradoxical between the approximate increases seen in OB-F$_{ab}$ *vs.* approximate depressive effects on maximal *in vitro* 3-$O$-MFPase activity. The elevated OB-F$_{ab}$ (per wet weight) after 67% V$_{O_2peak}$ but not after fatiguing exercise are first at odds with the reduction in the 3-$O$-MFPase activity (per wet weight) after fatiguing exercise. This divergence is consistent with the previous finding of a decline in muscle 3-$O$-MFPase activity with intense exercise but not in OB (Petersen et al., 2005) and with the well-described reduction in 3-$O$-MFPase activity with intense exercise (McKenna et al., 2008). Suggested factors contributing to this reduction in 3-$O$-MFPase activity independent of any effects on OB include elevations in reactive oxygen species, myoplasmic calcium (McKenna et al., 2008) and NKA subunit glutathionylation (Juel et al., 2015). Our findings are inconsistent when compared between results expressed per muscle wet weight *vs.* per protein. Whilst an increase (per protein) in OB-F$_{ab}$ was found after both 67% V$_{O_2peak}$ and after fatiguing exercise, there was a lack of effect (per protein) on 3-$O$-MFPase activity. A reduction in 3-$O$-MFPase activity (per protein) with exercise has been reported previously (Aughey et al., 2007; Fraser et al.,

2002), suggesting the reductions in activity per wet weight were not simply due to fluid shifts; it is unclear why no reduction was similarly found here. Others have reported that a decline in 3-$O$-MFPase activity with exercise was not replicated by a reduction in NKA activity when this was measured by Pi liberation (Juel et al., 2013). However, their subsequent paper did in fact report a decline in the latter measurement with exercise (Hostrup et al., 2014). Further detailed investigations into acute exercise effects on muscle NKA activity utilising different techniques are warranted. Nonetheless, it seems likely that acute exercise exerts diverging effects on NKA activity and content.

### Elevated serum digoxin was not associated with impaired K$^+$ homeostasis with exercise

Contrary to our hypothesis, elevated serum digoxin did not exacerbate plasma [K$^+$] with exercise in these healthy adults. Instead, [K$^+$] variables were remarkably consistent between DIG and CON, for both FF and LC exercise tests. Plasma [K$^+$]$_a$ during and following each of FF and LC exercise was unchanged by digoxin, whilst during FF, plasma [K$^+$]$_v$, [K$^+$]$_{a-v}$ difference and K$^+$ efflux into plasma across the contracting forearm muscles were also all unchanged by digoxin. In contrast, during LC, a lower [K$^+$]$_v$ and a greater [K$^+$]$_{a-v}$ were found with digoxin, indicating a greater net K$^+$ uptake by inactive forearm muscles. These small differences with digoxin are opposite to expected changes with NKA inhibition in inactive muscle and cannot readily be explained. We anticipated that digoxin would exacerbate the rise in plasma [K$^+$] with exercise, based on studies in patients with atrial fibrillation and heart failure, where 1.2–2.3 nmol l$^{-1}$ [digoxin] further increased [K$^+$] by ~0.2–0.3 mmol l$^{-1}$ during cycling exercise (Norgaard et al., 1991; Schmidt et al., 1995), increased the K$^+$ loss from the exercising leg by ~138% (Schmidt et al., 1995), and also in healthy adults, where digoxin elevated resting venous [K$^+$] (Edner et al., 1993). The non-impairment by digoxin of K$^+$ homeostasis with exhaustive dynamic exercise in these healthy participants is therefore clearly at odds with previous clinical studies (Norgaard et al., 1991; Schmidt et al., 1995). This was not due to inadequate systemic digitalisation, as [digoxin] was at expected clinical levels, at 0.8 nM. One possibility is that this [digoxin] was below a threshold required to perturb K$^+$ homeostasis during exercise; that is, 1.1–1.2 nM was previously shown to exert an effect (Norgaard et al., 1991; Schmidt et al., 1995). The lack of digoxin impairment of K$^+$ homeostasis is highly unlikely to be due to insufficient activation of NKA in contracting muscles, given the large disturbances to systemic [K$^+$] during both LC and FF exercise, the expected large increases in NKA activity with intense muscular contractions and the high *in vivo*

muscle NKA activity in both LC and FF trials, evidenced by the rapid post-exercise declines in $[K^+]_a$ and/or in $[K^+]_v$ after early and fatiguing exercise bouts. These rapid post-exercise reductions in systemic $[K^+]$ with intense exercise are probably consequent to decreased muscle interstitial $[K^+]$ induced by muscle NKA activation in the previously contracting musculature (Sejersted & Sjogaard, 2000; Lindinger & Cairns, 2021). The net $K^+$ uptake into forearm muscles during LC was also consistent with substantial *in vivo* NKA activity also in non-contracting muscles. Why this net $K^+$ uptake was elevated with DIG during LC is unclear. Catecholamines were not measured here to determine whether these might be elevated above placebo trials, which might have independently affected NKA activation and produced a counterbalancing effect to digoxin on $K^+$ homeostasis. However, this seems unlikely, given noradrenaline was reduced with digoxin in heart failure patients (van Veldhuisen et al., 1993). Furthermore, altered fluid shifts cannot explain the unchanged $[K^+]$ with digoxin, since arterio-venous $\Delta PV$ or $\Delta BV$ were unchanged by digoxin, in both FF or LC trials. Early studies in healthy males found that intravenous ouabain infusion of 0.05 mg min$^{-1}$ for 10 min did not affect forearm or hand blood flow (Glover et al., 1967). We found slightly lower forearm blood flow during FF with digoxin, but this seems unlikely to explain the lack of effect of digoxin on the $[K^+]_{a-v}$ and $K^+$ fluxes during FF. We were unable to obtain reliable forearm blood flow during leg exercise so cannot conclude whether any changes in perfusion might be related to the lower $[K^+]_v$ and elevated $[K^+]_{a-v}$ across the inactive forearm. Rather, these seem more likely to reflect a paradoxical greater NKA activation with digoxin during leg exercise. One possibility for the largely unchanged $[K^+]$ perturbations with exercise after digoxin is that intracellular $[Na^+]$ could have been slightly elevated due to digoxin inhibition of NKA, which would then increase activity of the remaining functional NKA, thereby counterbalancing inhibitory effects. Importantly, this study was conducted with healthy, young adults, whereas studies that reported exacerbated increases with digoxin in $[K^+]_a$ or $[K^+]_v$ in venous blood draining contracting muscles during exercise utilised clinical populations (Norgaard et al., 1991; Schmidt et al., 1995). Hence the impacts of digoxin therapy on $K^+$ regulation may be quite different in these populations. One previous finding in healthy individuals also reported no effect of acute digoxin on venous $[K^+]$ during handgrip exercise (Janssen et al., 2009). However, their findings are difficult to apply here, since their exercise intensity was low, induced only small increases in venous $[K^+]$ and $[Lac^-]$, and as it is unclear whether their blood sampling occurred during or after exercise, which markedly affects $[K^+]$; and also, since none of serum [digoxin], digoxin binding to muscle or forearm blood flow were measured.

The preservation of NKA in skeletal muscle after 14 days of oral digoxin in healthy humans is the most likely explanation for the remarkable consistency in $[K^+]$ observed between DIG and CON trials across several exercise modalities. Our findings are therefore consistent with $K^+$ being a tightly regulated physiological variable during and following exercise (McKenna et al., 2008) and in non-exercising conditions (McDonough & Youn, 2005). Although digoxin occupancy of NKA was not analysed for the forearm muscle, it is likely to be similar to quadriceps muscle, given these muscles all have a similar mixed fibre composition (Johnson et al., 1973). Whilst we cannot exclude the possibility that a higher serum [digoxin] and greater digoxin binding fraction in muscle are required to impair $K^+$ homeostasis in healthy individuals, such a study might be difficult to conduct due to the increased risks involved.

## Digoxin impaired skeletal muscle strength, but not exercise performance or fatiguability

An interesting finding was the reduced maximal voluntary strength of the quadriceps muscles with digoxin by 4%, measured across a range of limb velocities. Our research design in these 10 healthy participants featured careful control of test procedures, familiarisation and determination of within-subject variability for quadriceps strength and fatiguability, which were highly reproducible. This might explain why we detected a reduction whereas previous studies reported no change in muscle strength in only six, well-trained participants after digoxin (Sundqvist et al., 1983) or only tendency to reduced strength in only four participants (Bruce et al., 1968; Sundqvist et al., 1983). Others have reported that intra-arterial injection of high-dose ouabain enhanced electrically evoked muscle strength, whereas low-dose ouabain did not (Smulyan & Eich, 1976). The digoxin dose here was sufficient to induce small reductions in muscle strength, whereas a higher digoxin dosage might be required to induce more marked reductions in force, as demonstrated in isolated muscle preparations with inhibition by ouabain (Clausen, 2003). In genetically altered mice, reduced or absent NKA $\alpha_2$ isoform in muscle did not reduce evoked tetanic force in diaphragm (Radzyukevich et al., 2004), but did lower twitch and tetanic force in EDL muscle (Radzyukevich et al., 2013). This latter finding is unlikely to be applicable to our participants, however, due to the relatively small fraction of NKA bound by digoxin at this clinical serum concentration. Mice exhibiting lowered NKA $\alpha_1$ isoforms also had reduced mass and cross-sectional area in soleus muscle (Kutz et al., 2018). This suggests potential linkages between NKA, muscle mass and strength might also be present in human muscles. It is therefore interesting that a reduction in strength persisted with DIG even despite

preservation of functional muscle NKA. Another possibility is that these effects might also be via neural effects, linked to inhibition of neuronal NKA $\alpha_3$ isoforms, in the motor pathways and/or motor nerves, but this remains to be tested.

We hypothesised that fatiguability would be exacerbated with digoxin, due to NKA inhibition at rest and during exercise (Joreteg & Jogestrand, 1983), consistent with marked inhibitory effects of ouabain in isolated muscle preparations (Clausen, 2003) and with impaired treadmill running in mice with skeletal muscle-selective NKA $\alpha_2$ knockout (Radzyukevich et al., 2013). However, an important and consistent finding was the lack of effect of digoxin on fatiguability during three different exercise protocols, namely 50 repeated dynamic contractions of the quadriceps muscles and each of intense FF and LC exercise that concluded with a final exercise bout continued to fatigue. The quadriceps fatiguability and LC performance tests were all highly reproducible during variability trials, although a higher variability was evident for time to fatigue during FF. The lack of a digoxin effect on all these repeat contraction performance measures was internally consistent, suggesting variability was not the underlying cause for failure to detect an effect. The lack of effect of digoxin on torque during the fatigue test at 180° s$^{-1}$ probably reflects the typical variability, especially in initial efforts, in peak torque during these 50 repeated contractions (Fig. 5). Our findings were also consistent with a lack of effect of digoxin on isometric endurance during sustained handgrip contractions at 30% MVC (Bruce et al., 1968).

Elevated extracellular [K$^+$] at high concentrations can depress maximal force in rodent muscle preparations, but can also potentiate muscle force during twitch and submaximal contractions, thereby exerting dual effects (Pedersen et al., 2019). Thus, small increases in [K$^+$] with digoxin might actually benefit muscle performance during submaximal exercise, yet also contribute to fatiguability during fatiguing contractions when muscle activation increases. An important question is whether the peak [K$^+$] reached during exhaustive exercise (FF, ∼4.2 and ∼5.2 mM; LC, ∼6.5 and ∼5.0 mM, in arterial and antecubital venous plasma, respectively) were sufficiently high to produce muscular fatigue. Muscle interstitial [K$^+$] typically reaches ∼11–13 mM with exhaustive exercise and substantially exceeds arterial plasma [K$^+$], or plasma [K$^+$] in the venous effluent from contracting muscles (McKenna et al., 2008) with reported gradients of ∼4.3 and ∼3.8 mM (Nielsen et al., 2004) and of 6.5 and 5.8 mM (Green et al., 2000). It therefore seems likely that muscle interstitial [K$^+$] was similarly elevated 4–6 mM above our measured plasma values during these exhaustive contractions. If so, the interstitial [K$^+$] might have reached 9–11 mM during FF and 10.5–12.5 mM during LC and therefore at critical levels that may contribute to impaired

membrane excitability and muscle force reduction during intense contractions (McKenna et al., 2008). However, given the lack of changes in [K$^+$] after 14 days of digoxin, presumably linked to preservation of muscle functional NKA, we cannot determine here whether there was any effect of elevated [K$^+$] on fatigue.

The unchanged performance time and V$_{O_2peak}$ during two-legged cycling to exhaustion at 90% V$_{O_2peak}$ with DIG contrasts with findings of worsened incremental treadmill running time with digoxin in four healthy adults (Bruce et al., 1968), but is consistent with unchanged VO$_{2max}$ reported with a serum [digoxin] of ∼1.0 nmol l$^{-1}$ in well-trained (Sundqvist et al., 1983) and in untrained men (Russell & Reeves, 1963). Two of these studies were performed after short-term, higher dose digitalisation (Bruce et al., 1968; Russell & Reeves, 1963), also suggesting that the extent of digitalisation cannot explain the different findings. In contrast, digitalisation in heart failure patients, with serum [digoxin] of 1.1 nmol l$^{-1}$, increased peak exercise VO$_2$ by 16%, consistent with its myocardial inotropic effects (Sullivan et al., 1989). No effects of digoxin were found here during FF on O$_2$ uptake by the contracting forearm muscles. Together, these all point to no effect of digoxin on performance during intense exercise. However, any possible effect might not have been detected due to NKA being preserved in skeletal muscle in these participants.

## Conclusions

Oral digoxin administration for 14 days in healthy participants induced a typical clinical serum [digoxin] as well as a 7.6% skeletal muscle digoxin occupancy of NKA, as revealed by digoxin clearance with Digibind. However, digoxin did not depress the muscle [$^3$H]-ouabain binding site content, but rather, this was preserved at control levels, as was the maximal *in vitro* 3-O-MFPase activity. The total overall NKA content in resting muscle, as indicated by NKA bound by digoxin plus unbound NKA, was therefore increased with digoxin, which allowed functional NKA to be preserved. Whilst this suggests possible compensatory increase in NKA in resting skeletal muscle to protect against the expected functional depression in NKA with digoxin, we did not detect differences in OB+F$_{ab}$ between DIG and CON, which would be expected with NKA upregulation. The mechanism for this preservation therefore remains unresolved. Plasma K$^+$ dynamics during and following finger flexion and leg cycling exercise were also largely unperturbed by digoxin. The lack of change with digoxin in arterial K$^+$ disturbances during intense exercise is consistent with K$^+$ homeostasis being tightly regulated, with one important functional outcome being to ensure maintenance of muscle contractile function. Accordingly, digoxin also did not impair

fatigability during intense finger flexion, leg cycling exercise or repeated quadriceps contractions, although quadriceps muscle strength was reduced. The unchanged functional NKA in muscle with digoxin is consistent with the observed preservation of muscle function, fatiguability and $K^+$ regulation with exercise after digitalisation. These findings point to resilience of skeletal muscle NKA and function in healthy adults when undergoing a short digoxin challenge. Our findings in healthy adults also contrast with reports in heart failure and atrial fibrillation patients where digoxin reduced skeletal muscle NKA and exacerbated $[K^+]$ disturbances with exercise. A further surprising finding was that exercise caused a transient increase in muscle $[^3H]$-ouabain binding site content, although this varied somewhat across measures with and without Digibind and whether expressed per gram of wet weight or protein. We suggest that these findings may be due to a rapid combination of pre-existing NKA $\alpha$ and $\beta$ subunits to form new functional NKA $\alpha\beta$ complexes in muscle.

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

## Additional information

### Data availability statement

All data cited in the paper are available on the Open Science Forum and can be accessed at: https://osf.io/ku82x/?view_only=370e1f3f0da248bba9b6d28fc6a895d6.

### Competing interests

None.

### Author contributions

M.M., H.K., D.C.S. and R.S. conceived and designed the research. S.S., A.P., C.G., X.G., M.B., J.A., A.G., C.S., K.M., K.C., S.F., J.L., D.C.S., R.S. and M.M. conducted the experiments. S.S., A.P., X.G., K.M., K.C. and M.M. analysed the data. M.M., S.S., A.P. and X.G. performed the statistical analyses. M.M., S.S., A.P. and X.G. wrote the initial manuscript draft. All authors read and approved the final manuscript, except H.K. (deceased) and X.G. (departed university and lost contact) who each read earlier drafts.

### Funding

The study was funded by the National Health and Medical Research Council of Australia (No. 256603).

### Acknowledgements

We thank all participants for their generous contributions to this challenging study. We thank Professor Steve Selig, Dr Jason Bennie and Ms Bente Weidmann for assistance in some trials, Dr Chris Stathis for assistance with plasma and blood lactate analyses and Dr Matthew Lee, Dr Elizabeth Reisman and Ms Navabeh Zarekookandeh for assistance with Western blotting. We thank Dr Marina Skiba from Monash University for assistance with digoxin and ethics and Dr Sheue-Ching Oii from the Clinical Trial Pharmacy at Alfred Hospital. We acknowledge the expert contributions of our esteemed colleague Prof. Henry Krum who sadly passed away on 28 November 2015.

Open access publishing facilitated by Victoria University, as part of the Wiley – Victoria University agreement via the Council of Australian University Librarians.

### Keywords

digoxin, exercise, muscle strength, ouabain, potassium, skeletal muscle fatigue, sodium-potassium pump

## Supporting information

Additional supporting information can be found online in the Supporting Information section at the end of the HTML view of the article. Supporting information files available:

**Statistical Summary Document**
**Peer Review History**

