## [Peer Review History · The Journal of Physiology]

Oral digoxin effects on exercise performance, K⁺ regulation and skeletal muscle Na⁺,K⁺-ATPase in healthy humans

Simon Sostaric, Aaron Petersen, Craig A Goodman, Xiaofei Gong, Tai Juan Aw, Malcolm J Brown, Andrew Garnham, Collene H Steward, Kate T Murphy, Kate A Carey, James Leppik, Steve F Fraser, David Cameron-Smith, Henry Krum, Rod J Snow, and Michael J McKenna

DOI: 10.1113/JP283017

Corresponding author(s): Michael McKenna (michael.mckenna@vu.edu.au)

Review Timeline:

Submission Date:	01-Mar-2022
Editorial Decision:	13-Apr-2022
Revision Received:	12-May-2022
Editorial Decision:	01-Jun-2022
Revision Received:	22-Jun-2022
Editorial Decision:	04-Jul-2022
Revision Received:	05-Jul-2022
Accepted:	08-Jul-2022

Senior Editor: Scott Powers

Reviewing Editor: Bruno Grassi

Transaction Report:

Dear Professor McKenna,

Re: JP-RP-2022-283017 "Oral digoxin effects on K⁺ regulation with exercise, fatigue, skeletal muscle Na⁺,K⁺-ATPase and muscle strength in healthy humans" by Simon Sostaric, Aaron Petersen, Craig A Goodman, Xiaofei Gong, Tai Juan Aw, Malcolm J Brown, Andrew Garnham, Collene H Steward, Kate T Murphy, Kate A Carey, James Leppik, Steve F Fraser, David Cameron-Smith, Henry Krum, Rod J Snow, and Michael J McKenna

Thank you for submitting your manuscript to The Journal of Physiology. It has been assessed by a Reviewing Editor and by 1 expert Referees and I am pleased to tell you that it is considered to be acceptable for publication following satisfactory revision.

The reports are copied at the end of this email. Please address all of the points and incorporate all requested revisions, or explain in your Response to Referees why a change has not been made.

NEW POLICY: In order to improve the transparency of its peer review process The Journal of Physiology publishes online as supporting information the peer review history of all articles accepted for publication. Readers will have access to decision letters, including all Editors' comments and referee reports, for each version of the manuscript and any author responses to peer review comments. Referees can decide whether or not they wish to be named on the peer review history document.

Authors are asked to use The Journal's premium BioRender (<https://biorender.com/>) account to create/redraw their Abstract Figures. Information on how to access The Journal's premium BioRender account is here: <https://physoc.onlinelibrary.wiley.com/journal/14697793/biorender-access> and authors are expected to use this service. This will enable Authors to download high-resolution versions of their figures. The link provided should only be used for the purposes of this submission. Authors will be charged for figures created on this premium BioRender account if they are not related to this manuscript submission.

I hope you will find the comments helpful and have no difficulty returning your revisions within 4 weeks.

Your revised manuscript should be submitted online using the links in Author Tasks Link Not Available.

Any image files uploaded with the previous version are retained on the system. Please ensure you replace or remove all files that have been revised.

REVISION CHECKLIST:

- Article file, including any tables and figure legends, must be in an editable format (eg Word)
- Abstract figure file (see above)
- Statistical Summary Document
- Upload each figure as a separate high quality file
- Upload a full Response to Referees, including a response to any Senior and Reviewing Editor Comments;
- Upload a copy of the manuscript with the changes highlighted.

- A potential 'Cover Art' file for consideration as the Issue's cover image;
- Appropriate Supporting Information (Video, audio or data set https://jp.msubmit.net/cgi-bin/main.plex?form_type=display_requirements#supp).

To create your 'Response to Referees' copy all the reports, including any comments from the Senior and Reviewing Editors, into a Word, or similar, file and respond to each point in colour or CAPITALS and upload this when you submit your revision.

I look forward to receiving your revised submission.

If you have any queries please reply to this email and staff will be happy to assist.

Yours sincerely,

Scott K. Powers
Senior Editor
The Journal of Physiology
<https://jp.msubmit.net>
<http://jp.physoc.org>
The Physiological Society
Hodgkin Huxley House
30 Farringdon Lane
London, EC1R 3AW
UK
<http://www.physoc.org>
<http://journals.physoc.org>

REQUIRED ITEMS:

-Author photo and profile. First (or joint first) authors are asked to provide a short biography (no more than 100 words for one author or 150 words in total for joint first authors) and a portrait photograph. These should be uploaded and clearly labelled with the revised version of the manuscript. See Information for Authors for further details.

-You must start the Methods section with a paragraph headed Ethical Approval. If experiments were conducted on humans confirmation that informed consent was obtained, preferably in writing, that the studies conformed to the standards set by the latest revision of the Declaration of Helsinki, and that the procedures were approved by a properly constituted ethics committee, which should be named, must be included in the article file. If the research study was registered (clause 35 of the Declaration of Helsinki) the registration database should be indicated, otherwise the lack of registration should be noted as an exception (e.g. The study conformed to the standards set by the Declaration of Helsinki, except for registration in a database.). For further information see: <https://physoc.onlinelibrary.wiley.com/hub/human-experiments>

-Please upload separate high-quality figure files via the submission form.

-You must upload original, uncropped western blot/gel images (including controls) if they are not included in the manuscript. This is to confirm that no inappropriate, unethical or misleading image manipulation has occurred <https://physoc.onlinelibrary.wiley.com/hub/journal-policies#imagmanip> These should be uploaded as 'Supporting information for review process only'. Please label/highlight the original gels so that we can clearly see which sections/lanes have been used in the manuscript figures.

-A Statistical Summary Document, summarising the statistics presented in the manuscript, is required upon revision. It must be on the Journal's template, which can be downloaded from the link in the Statistical Summary Document section here: https://jp.msubmit.net/cgi-bin/main.plex?form_type=display_requirements#statistics

-Papers must comply with the Statistics Policy https://jp.msubmit.net/cgi-bin/main.plex?form_type=display_requirements#statistics

In summary:

-If n {less than or equal to} 30, all data points must be plotted in the figure in a way that reveals their range and distribution. A bar graph with data points overlaid, a box and whisker plot or a violin plot (preferably with data points included) are acceptable formats.

-If $n > 30$, then the entire raw dataset must be made available either as supporting information, or hosted on a not-for-profit repository e.g. FigShare, with access details provided in the manuscript.

-' n ' clearly defined (e.g. x cells from y slices in z animals) in the Methods. Authors should be mindful of pseudoreplication.

-All relevant ' n ' values must be clearly stated in the main text, figures and tables, and the Statistical Summary Document

(required upon revision)

-The most appropriate summary statistic (e.g. mean or median and standard deviation) must be used. Standard Error of the Mean (SEM) alone is not permitted.

-Exact p values must be stated. Authors must not use 'greater than' or 'less than'. Exact p values must be stated to three significant figures even when 'no statistical significance' is claimed.

-Statistics Summary Document completed appropriately upon revision

-Please include an Abstract Figure. The Abstract Figure is a piece of artwork designed to give readers an immediate understanding of the research and should summarise the main conclusions. If possible, the image should be easily 'readable' from left to right or top to bottom. It should show the physiological relevance of the manuscript so readers can assess the importance and content of its findings. Abstract Figures should not merely recapitulate other figures in the manuscript. Please try to keep the diagram as simple as possible and without superfluous information that may distract from the main conclusion(s). Abstract Figures must be provided by authors no later than the revised manuscript stage and should be uploaded as a separate file during online submission labelled as File Type 'Abstract Figure'. Please ensure that you include the figure legend in the main article file. All Abstract Figures should be created using BioRender. Authors should use The Journal's premium BioRender account to export high-resolution images. Details on how to use and access the premium account are included as part of this email.

EDITOR COMMENTS

Reviewing Editor:

Comments for Authors to ensure the paper complies with the Statistics Policy :
Precise P values should be given. $P < 0.05$ is not acceptable.

Comments to the Author:

The manuscript is the result of a randomized cross-over trial, in which young healthy subjects have been treated with digoxin and placebo to investigate the effects on Na-K pump content and activity, K homeostasis, performance and fatigability in various exercise tests. The topic is of interest and the data may have impact in the field. The manuscript derives from the merging of two studies previously presented as companion papers, and rejected as such by the Journal. The manuscript is of interest, but it is too long and with too many hypotheses. The referee does a nice job in suggesting to the authors several sections which could be deleted without compromising the main message of the study. A much more focussed (and more concise) manuscript should derive by this intervention, which is strongly recommended.

Senior Editor:

Comments for Authors to ensure the paper complies with the Statistics Policy :

This paper is an interesting case where displaying raw data points would likely confuse interpretation by readers (due to the nature of the study). Nonetheless, I will email the RE to gain their perspective on the data presentation prior to acceptance of the paper.

If the statistical summary document has errors please describe what is incorrect. :

No statistical summary provided

Comments to the Author:

Thank you for submitting your work to the Journal of Physiology. Your report has undergone a thorough evaluation by both a review editor (RE) and an expert referee. Both the RE and referee agree that your work is interesting and provides novel insight into the topic investigated. Nonetheless, the reviewer and RE have provided several suggestions to improve the final product of this work. We look forward to receiving your revised manuscript.

REFEREE COMMENTS

Referee #2:

The study by Sostaric et al. reports a randomized cross-over trial, where young healthy persons have been treated with digoxin and placebo to investigate the effects on Na-K pump content and activity, K homeostasis, performance and fatigability in various exercise tests. The manuscript covers an impressive amount of work and data which is generally well analyzed and reported. I think the rationale for the main questions is sound and the experiment is well designed and measurements are very well performed with use of important validation procedures, reproducibility tests and familiarization

trials. However, the study aims to investigate no less than 13 hypotheses based on the trial, which leads to problems as described below.

Major points:

The overwhelming number of hypotheses addressed precludes a clear focus throughout the paper and leads to an excessively long manuscript. In the following are some comments and suggestions for revising the paper with the goal of presenting the main parts of the trial in a more concise form.

I think that all data on proteins and mRNA of NKA isoforms could be omitted. The protein data are by the authors own admission of questionable quality due to low sample size and variability of the assay (discussion p 26) and mRNA data also suffer from huge unexplained variations (table 1) and they do not represent any consistent findings in themselves,

nor do they add important information to the remaining hypotheses.

I don't think the comparison between exercise tests (finger flexion and leg cycling) is relevant. The information derived from this analysis has nothing to do with the main question (digoxin effects) and the shown differences in K movements could already be deduced from previous studies. The two exercise tests could still be presented, to support the finding of no increase in fatiguability, but without the comparison.

I would also suggest omitting the data on Lac and H⁺. These data are very indirectly related to the main question. Since there was no effect of digoxin on fatiguability and K⁺ homeostasis, it makes sense that Lac⁻ and H⁺ also show no consistent changes, but I don't see how reporting this strengthens the main story-line

Finally, I think the acute effects of exercise on NKA pump content also seems like a separate study. It is not clear why 4 biopsies were made over the exercise period. The introduction states that the hypothesis was "no acute effects". Now it turned out that there were some exercise-induced effects, but these were unrelated to digoxin and could be reported elsewhere.

With the above omissions, the report would still include the central measurements of performance and fatiguability in three exercise tests, K homeostasis during exercise, NKA content (ouabain binding) and NKA function (3OMFPase activity) and blood digoxin concentrations and would answer the central hypotheses regarding the effects of digoxin on NKA, performance and K-transport. I think the manuscript would be more clear and focused if it was cut down to these central outcomes.

Minor:

Title needs revision. Too many outcomes mentioned. Perhaps use the word "performance", which could encompass both strength and fatigue measurements

Abstract is confusing, with many outcomes crammed into few sentences. Report only the main outcomes in this section.

Introduction:

Why was there a hypothesis that digoxin would reduce strength? (hypothesis number 10). During a brief MVC, there would probably not be time enough for build up of extracellular K, so wouldn't the expectation be that strength (brief effort) was maintained, but endurance (lasting effort) would be depressed with digoxin?

Methods:

Muscle strength measurements: Were three attempts made for all contraction speeds? It is stated that peak torque was used for FV curve. Was this peak torque of the best attempt or average peak from all three attempts?

The control experiments demonstrating the validity of the digoxin Fab method are very nice. However, in the experiment concerning the possible effect of time, only the P-value is reported. Burt a P-value (3 vs 3) is not very informative. Report instead the mean values and perhaps confidence intervals.

Results:

Could it perhaps be interesting to calculate and report the 3OMFPase values relative to the measured NKA content?

Discussion

The first section of the discussion does not presently re-address all the hypotheses from the introduction. This problem could be solved by reducing the number of hypotheses.

END OF COMMENTS

Confidential Review

01-Mar-2022

Author Responses to Senior Editor, Reviewing Editor and Referee#2

10th May, 2022

JP-RP-2022-283017 "Oral digoxin effects on K⁺ regulation with exercise, fatigue, skeletal muscle Na⁺,K⁺-ATPase and muscle strength in healthy humans" by Sostaric, Petersen et al.

Critique of JP-RP-2022-283017 and Author Responses

Senior Editor: Thank you for submitting your manuscript to The Journal of Physiology. It has been assessed by a Reviewing Editor and by 1 expert Referees and I am pleased to tell you that it is considered to be acceptable for publication following satisfactory revision

Response: Thank you we are delighted with this outcome. We confirm that all requested changes have been made to the ms as detailed below.

REQUIRED ITEMS:

-Author photo and profile

Response: We have included Author photo and profile (for joint first authors)

-You must start the Methods section with a paragraph headed Ethical Approval..... If the research study was registered (clause 35 of the Declaration of Helsinki) the registration database should be indicated, otherwise the lack of registration should be noted as an exception.

Response: We have included Ethical Approval as initial section in Methods

-A Statistical Summary Document, summarising the statistics presented in the manuscript, is required upon revision.

Response: Statistical Summary document has been included.

-Papers must comply with the Statistics Policy

Response: we have now amended P values as indicated below. We confirm that the ms now complies with the Statistics Policy.

-Please include an Abstract Figure.

Response: We have included the Abstract Figure.

Reviewing Editor: Precise P values should be given. P<0.05 is not acceptable

Response: In the ms we reported statistical significance on numerous occasions as P<0.001, since IBM SPSS Statistics 27, which we used to perform

statistical analyses, does not report exact P values at levels below $P < 0.001$. To comply with Journal of Physiology requirements we have now amended these instances to report as $P = 0.001$. We have accordingly included in the methods statistics section: “since this package does not report exact P values at lower than $P < 0.001$, for these data they are reported here as $P = 0.001$.” The figures still retain $P <$ for symbols where differences encompass a range of times, but caption states the exact P values for these times are stated in the results text.

Reviewing Editor: The manuscript is the result of a randomized cross-over trial, in which young healthy subjects have been treated with digoxin and placebo to investigate the effects on Na-K pump content and activity, K homeostasis, performance and fatiguability in various exercise tests. The topic is of interest and the data may have impact in the field. The manuscript derives from the merging of two studies previously presented as companion papers, and rejected as such by the Journal. The manuscript is of interest, but it is too long and with too many hypotheses. The referee does a nice job in suggesting to the authors several sections which could be deleted without compromising the main message of the study. A much more focussed (and more concise) manuscript should derive by this intervention, which is strongly recommended.

Response: We have amended the ms making it much shorter and more focussed. In brief, we have reduced the hypotheses from 13 to 6 and deleted all but one of the sections requested by the referee. These major changes have resulted in a reduction of the ms by ~8 pages of text from each of Introduction, Methods, Results and Discussion; plus a reduction from 4 to 2 Tables and from 14 to 7 figures, as detailed below.

Senior Editor: This paper is an interesting case where displaying raw data points would likely confuse interpretation by readers (due to the nature of the study). Nonetheless, I will email the RE to gain their perspective on the data presentation prior to acceptance of the paper..

Response: We have responded to the above request from Review Editor to include specific P values.

Senior Editor: No statistical summary provided.

Response: A statistical summary has now been provided.

Senior Editor: Thank you for submitting your work to the Journal of Physiology. Your report has undergone a thorough evaluation by both a review editor (RE) and an expert referee. Both the RE and referee agree that your work is interesting and provides novel insight into the topic investigated. Nonetheless, the reviewer and RE have provided several suggestions to improve the final product of this work. We look forward to receiving your revised manuscript.

Response: Thank you. We have responded to all requests and amended the ms accordingly, as detailed below.

Referee#2 Critique and Author Response

Referee#2: The study by Sostaric et al. reports a randomized cross-over trial, where young healthy persons have been treated with digoxin and placebo to investigate the effects on Na-K pump content and activity, K homeostasis, performance and fatiguability in various exercise tests. The manuscript covers an impressive amount of work and data which is generally well analyzed and reported. I think the rationale for the main questions is sound and the experiment is well designed and measurements are very well performed with use of important validation procedures, reproducibility tests and familiarization trials. However, the study aims to investigate no less than 13 hypotheses based on the trial, which leads to problems as described below.

Response: We thank the reviewer for their thorough analysis of the ms and their positive comments.

Referee#2:

Major points:

The overwhelming number of hypotheses addressed precludes a clear focus throughout the paper and leads to an excessively long manuscript. In the following are some comments and suggestions for revising the paper with the goal of presenting the main parts of the trial in a more concise form.

Response: We have reduced the hypotheses from 13 to 6 and deleted the relevant sections requested by the referee, except one, which we argue against below. These major changes have resulted in a major reduction of the ms, by ~8 pages of text from each of Introduction, Methods, Results and Discussion; plus a reduction from 4 to 2 Tables and from 14 to 7 figures. This achieves the request to produce a more concise ms.

Referee#2: I think that all data on proteins and mRNA of NKA isoforms could be omitted. The protein data are by the authors own admission of questionable quality due to low sample size and variability of the assay (discussion p 26) and mRNA data also suffer from huge unexplained variations (table 1) and they do not represent any consistent findings in themselves, nor do they add important information to the remaining hypotheses.

Response: The muscle protein and mRNA data and all methods, results and discussion have now been omitted from the ms. This removes one Figure and 2 Tables from the ms. We have moved these Tables in the supplementary material.

Referee#2: I don't think the comparison between exercise tests (finger flexion and leg cycling) is relevant. The information derived from this analysis has nothing to do with the main question (digoxin effects) and the shown differences in K movements could already be deduced from previous studies. The two exercise tests could still be presented, to support the finding of no increase in fatiguability, but without the comparison.

Response: We have omitted this comparison, and removed each of the hypothesis, the results and discussion.

Referee#2: I would also suggest omitting the data on Lac and H⁺. These data are very indirectly related to the main question. Since there was no effect of

digoxin on fatiguability and K⁺ homeostasis, it makes sense that Lac⁻ and H⁺ also show no consistent changes, but I don't see how reporting this strengthens the main story-line.

Response: We have omitted all of the Lac⁻ and H⁺ data from the ms and therefore also removed the relevant hypotheses, sections in methods and discussion. This removes 6 Figures and 2/3 of one Table. We have moved the data into the supplementary material.

Referee#2: Finally, I think the acute effects of exercise on NKA pump content also seems like a separate study. It is not clear why 4 biopsies were made over the exercise period. The introduction states that the hypothesis was "no acute effects". Now it turned out that there were some exercise-induced effects, but these were unrelated to digoxin and could be reported elsewhere.

Response: The exercise measures of NKA content are in fact an important part of the study, not separate.

The four biopsies were included for two main reasons:

- a) to detect potential acute increases in NKA isoform mRNA and protein after exercise and whether these might then be modified by digoxin. The rationale for this was explained in the original ms, however, all NKA isoform mRNA and protein data are now deleted as requested.
- b) to determine any interactive impacts of exercise plus digoxin on muscle NKA binding of digoxin. We did include a rationale for this in the original ms introduction, referring in two paragraphs to an apparent effect of exercise causing increased digoxin binding to skeletal muscle and a decline in serum digoxin during exercise (Joretteg and Jogestrand 1983). The initial reference on Page 6, para 1 has been slightly amended to clarify this point. The second very similar statement was in the subsequent paragraph which referred to genes and has now been deleted. We have also included the following statement in the methods muscle biopsy procedures section to further clarify this: "The exercise and recovery biopsies were included to detect possible digoxin effects on NKA-digoxin binding influenced by exercise."

Our findings demonstrated a lack of effect of digoxin on muscle OB, which clearly differs to previous findings of increased digoxin binding in muscle (Joretteg and Jogestrand 1983). This might be due to the higher dosage of digoxin taken in that study, double that in ours (0.5 vs 0.25 mg/d x 14 d), and with a resting serum [digoxin] of 1.4-1.5 nM, which was ~75-88% higher than the 0.8 nM found in our study. We argue that this different finding alone warrants retention of the exercise and recovery muscle biopsy OB data. Retention of the exercise OB data for this purpose then also requires explanation of the acute exercise effects on muscle OB. These findings of an increase in OB with acute exercise are novel and we therefore also argue also for their retention on this basis. As these findings are integral to the main study it makes most sense to include these in the current ms than elsewhere.

We have inserted the following text into the discussion Page 29, para 3): "In that study however, the digoxin dosage taken was double that of the present

study (i.e. 0.5 vs 0.25 mg.d-1, for 14 d) and the serum [digoxin] achieved was 75-88% higher than the in our study (i.e. 1.4-1.5 vs 0.8 nM). Whether this explains the discrepancy between the two studies on muscle NKA content is unclear.”

No rationale was given for removal of this data other than thought to be unrelated to digoxin; we trust that we have now clearly demonstrated the importance of both digoxin plus acute exercise in the study.

We further note that our removal of substantial data sets from the ms results (plasma Lac-, H+, muscle NKA isoform mRNA and muscle NKA isoform protein), reduction in hypotheses, as well as corresponding text from introduction, methods and discussion now substantially reduces the ms size and sharpens the focus on muscle NKA, K+ and performance. This also then makes removal of the exercise effects on OB less important as a means to reduce ms size. Therefore the exercise and NKA content data were retained in the ms.

Referee#2: With the above omissions, the report would still include the central measurements of performance and fatiguability in three exercise tests, K homeostasis during exercise, NKA content (ouabain binding) and NKA function (3OMFPase activity) and blood digoxin concentrations and would answer the central hypotheses regarding the effects of digoxin on NKA, performance and K-transport. I think the manuscript would be more clear and focused if it was cut down to these central outcomes.

Response: We agree and have made all changes requested, except for retention of the muscle NKA OB data during exercise and recovery.

Referee#2:

Minor:

Title needs revision. Too many outcomes mentioned. Perhaps use the word "performance", which could encompass both strength and fatigue measurements

Response: The title has been amended, with inclusion of performance, removal of strength and isoforms, to now read: “Oral digoxin effects on exercise performance, K+ regulation and skeletal muscle Na+,K+-ATPase in healthy humans”

Referee#2: Abstract is confusing, with many outcomes crammed into few sentences. Report only the main outcomes in this section

Response: The abstract has been substantially revised to focus on main outcomes, with exclusion of all muscle isoform mRNA and protein, plasma [Lac-] and [H+] data and with simplification of the exercise [K+] data.

Referee#2: Introduction: Why was there a hypothesis that digoxin would reduce strength? (hypothesis number 10). During a brief MVC, there would

probably not be time enough for build up of extracellular K, so wouldn't the expectation be that strength (brief effort) was maintained, but endurance (lasting effort) would be depressed with digoxin?

Response: Our hypothesis was based on apparent trends in earlier papers, with the mechanism probably not relating to K⁺ build up, but rather to possible activation of NKA signalling pathways in muscle. This hypothesis has however, now been deleted from the ms.

Referee#2: Methods:

Muscle strength measurements: Were three attempts made for all contraction speeds? It is stated that peak torque was used for FV curve. Was this peak torque of the best attempt or average peak from all three attempts?

Response: Three attempts were made at each contraction speed and the highest peak torque of these three was utilised as the peak torque. The ms methods have been slightly revised to clarify these points (Page 8, para 3).

Referee#2: The control experiments demonstrating the validity of the digoxin Fab method are very nice. However, in the experiment concerning the possible effect of time, only the P-value is reported. Burt a P-value (3 vs 3) is not very informative. Report instead the mean values and perhaps confidence intervals.

Response: We have now inserted the raw data as well as the mean(SD).

Referee#2: Results: Could it perhaps be interesting to calculate and report the 3OMFPase values relative to the measured NKA content?

Response: We have calculated this ratio and inserted data into results (Page 17).

The calculated ratio of 3-O-MFPase activity/ouabain binding content was reduced from rest at both 67% VO_{2peak} (P=0.008) and fatigue (P=0.003) and recovered at 3 h post-exercise. The ratio of 3-O-MFPase activity/ouabain binding content did not differ between DIG and CON (P=0.770).

The ratio was identical whether calculated per wet weight or per protein, since this calculation effectively cancels out the protein content.

We avoided discussion of possible exercise or fatigue effects on 3-O-MFPase activity based on reasons outlined in the methods (Page 15). Therefore we have not discussed this data in the ms. This is consistent with the lack of effect of digoxin on the calculated ratio and also the request by the reviewer to reduce the overall size of the ms.

Referee#2: Discussion. The first section of the discussion does not presently re-address all the hypotheses from the introduction. This problem could be solved by reducing the number of hypotheses.

Response: We have accordingly removed the hypotheses not referred to here, regarding Lac- and H+.

Dear Professor McKenna,

Re: JP-RP-2022-283017R1 "Oral digoxin effects on exercise performance, K⁺ regulation and skeletal muscle Na⁺,K⁺-ATPase in healthy humans" by Simon Sostaric, Aaron Petersen, Craig A Goodman, Xiaofei Gong, Tai Juan Aw, Malcolm J Brown, Andrew Garnham, Collene H Steward, Kate T Murphy, Kate A Carey, James Leppik, Steve F Fraser, David Cameron-Smith, Henry Krum, Rod J Snow, and Michael J McKenna

Thank you for submitting your revised Research Article to The Journal of Physiology. It has been assessed by the original Reviewing Editor and Referees and has been well received. Some final revisions have been requested.

The reports are copied at the end of this email. Please address all of the points and incorporate all requested revisions, or explain in your Response to Referees why a change has not been made.

NEW POLICY: In order to improve the transparency of its peer review process The Journal of Physiology publishes online as supporting information the peer review history of all articles accepted for publication. Readers will have access to decision letters, including all Editors' comments and referee reports, for each version of the manuscript and any author responses to peer review comments. Referees can decide whether or not they wish to be named on the peer review history document.

Authors are asked to use The Journal's premium BioRender (<https://biorender.com/>) account to create/redraw their Abstract Figures. Information on how to access The Journal's premium BioRender account is here: <https://physoc.onlinelibrary.wiley.com/journal/14697793/biorender-access> and authors are expected to use this service. This will enable Authors to download high-resolution versions of their figures. The link provided should only be used for the purposes of this submission. Authors will be charged for figures created on this premium BioRender account if they are not related to this manuscript submission.

I hope you will find the comments helpful and have no difficulty returning your revisions within 2 weeks.

Your revised manuscript should be submitted online using the links in Author Tasks Link Not Available.

Any image files uploaded with the previous version are retained on the system. Please ensure you replace or remove all files that have been revised.

REVISION CHECKLIST:

- Article file, including any tables and figure legends, must be in an editable format (eg Word)
- Abstract figure file (see above)
- Statistical Summary Document
- Upload each figure as a separate high quality file
- Upload a full Response to Referees, including a response to any Senior and Reviewing Editor Comments;
- Upload a copy of the manuscript with the changes highlighted.

- A potential 'Cover Art' file for consideration as the Issue's cover image;
- Appropriate Supporting Information (Video, audio or data set https://jp.msubmit.net/cgi-bin/main.plex?form_type=display_requirements#supp).

To create your 'Response to Referees' copy all the reports, including any comments from the Senior and Reviewing Editors, into a Word, or similar, file and respond to each point in colour or CAPITALS and upload this when you submit your revision.

I look forward to receiving your revised submission.

If you have any queries please reply to this email and staff will be happy to assist.

Yours sincerely,

Scott K. Powers
Senior Editor
The Journal of Physiology
<https://jp.msubmit.net>
<http://jp.physoc.org>
The Physiological Society
Hodgkin Huxley House
30 Farringdon Lane
London, EC1R 3AW
UK
<http://www.physoc.org>
<http://journals.physoc.org>

REQUIRED ITEMS:

-Papers must comply with the Statistics Policy https://jp.msubmit.net/cgi-bin/main.plex?form_type=display_requirements#statistics

In summary:

-If $n \leq 30$, all data points must be plotted in the figure in a way that reveals their range and distribution. A bar graph with data points overlaid, a box and whisker plot or a violin plot (preferably with data points included) are acceptable formats.

-If $n > 30$, then the entire raw dataset must be made available either as supporting information, or hosted on a not-for-profit repository e.g. FigShare, with access details provided in the manuscript.

- n clearly defined (e.g. x cells from y slices in z animals) in the Methods. Authors should be mindful of pseudoreplication.

-All relevant n values must be clearly stated in the main text, figures and tables, and the Statistical Summary Document (required upon revision)

-The most appropriate summary statistic (e.g. mean or median and standard deviation) must be used. Standard Error of the Mean (SEM) alone is not permitted.

-Exact p values must be stated. Authors must not use 'greater than' or 'less than'. Exact p values must be stated to three significant figures even when 'no statistical significance' is claimed.

-Statistics Summary Document completed appropriately upon revision

EDITOR COMMENTS

Reviewing Editor:

Reviewer 2 still has a few comments/suggestions, which should be taken in consideration by the authors.

Senior Editor:

Comments for Authors to ensure the paper complies with the Statistics Policy:
Please provide raw data points in histograms as required by the journal.

Comments to the Author:

Thank you for submitting the revision of your report. Referee #1 is satisfied by your revision; however, both the review editor and referee #2 have requested a few additional (minor) revisions prior to acceptance.

REFEREE COMMENTS

Referee #2:

The Authors have responded adequately to the comments given initially, and I think the manuscript is more clear in the present form.

I have few more comments:

In the results section the text mentions many single time points comparisons giving relative changes and p values. I think the figures nicely shows these particular changes or differences and would suggest that the text is shortened and rather focus on describing the patterns and only single out the most important specific changes.

Consider moving the section about plasma volume, Hct, BV and Hb to a supplementary section. These measurements are of course important to ascertain that fluid shifts do not affect the observations, but this can be simplified to few sentences and the results be made available further scrutiny in a supplementary file, perhaps in the form of a table.

There is a discussion about the validity of the 3-O-MFPase assay in the methods section. It reads like the authors are a bit uncertain about this method. The finding presented in the results section is a bit paradoxical with respect to the changes during exercise. 3-O-MFPase activity goes down, while ouabain binding goes up. This result is now reported in the results, but is not discussed. I think it would be relevant to discuss this paradoxical finding - or alternatively, if the authors do not believe in the validity of the 3-O-MFPASE measurement, then simply omit these results.

Methods p5 still includes a section on the NKA subunit isoforms. I don't think this is necessary, when these measurements have been taken out of the manuscript

Discussion p 26. The section on what level of K⁺ changes can be expected in the interstitium seems a bit speculative. This part could be shortened or omitted.

The digoxin occupancy is reported to be around 8%, but when you calculate the difference in digoxin occupancy between PLA and DIG it is actually only a 4% higher occupancy in DIG. Perhaps it could be discussed if this very small difference is a possible explanation for the lack of changes seen in K levels and K fluxes between the two interventions.

END OF COMMENTS

1st Confidential Review

12-May-2022

JP-RP-2022-283017 "Oral digoxin effects on K⁺ regulation with exercise, fatigue, skeletal muscle Na⁺,K⁺-ATPase and muscle strength in healthy humans" by Sostaric, Petersen et al.

Critique of JP-RP-2022-283017 and Author Responses

Senior Editor: Thank you for submitting your revised Research Article to The Journal of Physiology. It has been assessed by the original Reviewing Editor and Referees and has been well received. Some final revisions have been requested.

Response: Thank you these have each been addressed below.

REQUIRED ITEMS:

Papers must comply with the Statistics Policy https://jp.msubmit.net/cgi-bin/main.plex?form_type=display_requirements#statistics

Response: This point is addressed below and our ms complies with the stats policy.

Reviewing Editor: Reviewer 2 still has a few comments/suggestions, which should be taken in consideration by the authors.

Response: These have each been considered and responded to below

Senior Editor: Comments for Authors to ensure the paper complies with the Statistics Policy:

Please provide raw data points in histograms as required by the journal.

Response:

1. Raw Data Presentation.

We argue that regarding presentation of raw data points, our ms already complies with the Journal of Physiology Statistics Policy, as detailed below, noting that our data is both time course and well in excess of n=30 measurement points. Therefore no changes were made to graphical presentation. The Policy states:

"If $n \leq 30$, all data points must be plotted in the figure in a way that reveals their range and distribution. A bar graph with data points overlaid, a box and whisker plot or a violin plot (the latter two also preferably with data points included) are acceptable formats. Note: if each subject has numerous data points associated with it (e.g. time course data), we would treat 'n' as being each data point, not the number of subjects."

"If $n > 30$, data points do not need be plotted in the figure but the entire raw dataset must be uploaded either as 'Supporting Information for online publication' (which will be published online with the article) or hosted on a not-for-profit repository e.g. FigShare, with access details provided in the manuscript."

All data presented graphically in the ms greatly exceeds $n > 30$, where 'n' as being each data point, not the number of subjects; As such this data presentation already complies with the policy. We have also included all raw data in Open Science Forum.

We did nonetheless already include raw data points in each of Figures 2-4 for the muscle OB and 3-O-MFPase data, even though final data point numbers exceed this ($n = 76-77$ data points), as this data set size was manageable even with presentation using bar charts.

However, the data set for Figures 5-7 far exceed this, all between $n > 200$ and $n = 1000$. As such, the data is considerably easier to understand when presented as a time series. We therefore argue that to change this

presentation to comply with Senior Editor's request is not required by the journal stats policy and also will make this much more complex and therefore less understandable. No changes were therefore made to this graphical presentation.

2. Measurement data point numbers.

Details of numbers of data points (n) for these Figures 2-7 are as follows:

Figure 2, OB-F_{ab}, final n = 76 data points (4 measurement times x 2 trials x 10 participants, minus 4 missing data points)

Figure 3, OB+F_{ab}, final n = 76 data points (4 measurement times x 2 trials x 10 participants, minus 4 missing data points)

Figure 4, 3-O-MFPase activity, final n = 77 data points (4 x 2 trials x 10 participants, minus 3 missing data points)

Figure 5, Cybex peak torque and fatigue:

Fig5A, peak torque-velocity, final n = 138 data points (7 measurement velocities x 10 participants x 2 trials, minus 2 missing data points)

Fig 5B, peak torque-fatigue, final n = 1,000 data points (50 measurement times (contractions) x 10 participants x 2 trials)

Figure 6, [K⁺] during finger flexion exercise:

Fig 6A, arterial [K⁺], final n = 242 data points (14 measurement times x 2 trials x 9 participants = 252 data points, minus 10 missing data points);

Fig 6B, venous [K⁺], final n= 218 data points (14 measurement times x 2 trials x 8 participants = 224 data points, minus 6 missing data points);

Fig 6C, [K⁺]_{a-v differences} (uncorrected), final n= 201 data points (14 measurement times x 2 trials x 8 participants= 224 data points, minus 23 missing data points; NB for one participant, 7 data points were missing for one trial, but data sets were matched, with time matching data therefore excluded for that individual in both of the two trials).

Figure 7, [K⁺] during leg cycling exercise:

Fig 7A, arterial [K⁺], final n= 218 data points (n = 14 measurement times x 2 trials x 10 participants = 280 data points, minus 6 missing data points);

Fig 7B, venous [K⁺], final n= 216 data points (n= 14 measurement times x 2 trials x 8 participants = 224 data points, minus 8 missing data points);

Fig 7C, [K⁺]_{a-v difference} (uncorrected), final n= 209 data points (n= 14 measurement times x 2 trials x 8 participants= 224 data points, minus 15 missing data points, including for the one participant with time matched exclusions as for venous [K⁺])

For data in Table 2, K⁺ Flux data during FF, the exact number of data measurement points was n= 168 data points (13 measurement times x 2 trials x 7 participants= 182 data points, minus 14 missing data points; NB for one participant, 6 data points were missing for one trial, but data sets were matched, with time matching data therefore excluded for that individual in both of the two trials).

3. Inclusion of Measurement Data numbers to Comply with Policy.

However, we realised that we did not fully comply with the policy on stating exact 'n' using the definition appropriate to our data and to adhere, we have now added the above final "n" for each measurement variable in the Statistics section in the main text as well as in the Figure legends.

"n' values. All 'n' values must be clearly stated main text, figures and their legends or tables and the Statistical Summary Document with 'n' clearly defined (e.g. x cells from y slices in z animals) in each location. Authors should be mindful of how 'n' is defined to avoid pseudoreplication."

New text has added to the ms in the Methods statistics section as stated below and appropriately abbreviated also in the corresponding Figure captions. We also corrected an error in the Methods text where we incorrectly stated n=10 for FF time to fatigue, whereas n=9 participants is correct for this measure (this was correctly stated as n=9 in the Table 1 caption).

New text added in Methods Statistics for compliance with policy:

"Exact number of data measurement points, defined as number of participants x number of trials x number of measurements minus missing data points), was 76, 76 and 77 for OB-F_{ab}, OB-F_{ab} and 3-O-MFPase activity, respectively."

"Exact number of data measurement points for torque-velocity was 138 and 1000 for fatigue data."

"Sample size for functional tests was n=10 for Cybex tests and leg cycling exercise performance and n=9 for finger flexion exercise performance. Exact number of data measurement points for Cybex tests was 138 for torque-velocity and 1000 for fatigue data and for time to fatigue during FF was 18 and LC was 20."

"Exact number of measurement points during FF for [K⁺]_a, [K⁺]_v, [K⁺]_{a-v}, was 242, 218 and 201, respectively."

"Exact number of measurement points during LC for [K⁺]_a, [K⁺]_v, [K⁺]_{a-v}, was 218, 216 and 209, respectively."

"For calculated K⁺ flux data during FF, the exact number of data measurement points was 168."

Senior Editor: Thank you for submitting the revision of your report. Referee #1 is satisfied by your revision; however, both the review editor and referee #2 have requested a few additional (minor) revisions prior to acceptance.

Response: We have responded to the Review Editor above; Referee#2 comments have each been responded to below.

Referee#2 Critique and Author Response

Referee#2: In the results section the text mentions many single time points comparisons giving relative changes and p values. I think the figures nicely shows these particular changes or differences and would suggest that the text is shortened and rather focus on describing the patterns and only single out the most important specific changes.

Response:

Our data set is complex, with for example, the [K⁺] data presented in both Figure 6 (Finger Flexion [K⁺]) and Figure 7 (Leg Cycling [K⁺]) each having 14 serial time points as well as two groups (DIG, CON) for comparisons. Furthermore in Figure 6, five samples were taken during different exercise bouts, three samples between exercise bouts and a further five in recovery; in Figure 7 six samples were taken during different exercise bouts, two samples between exercise bouts and five in recovery. This makes description of [K⁺] changes during these time points/groups quite complex.

In original submitted version we did try to simplify the results text almost exactly as suggested by the reviewer,

grouping similar responses together to describe patterns by e.g. stating where appropriate $P < 0.05$, or $P < 0.01$ so that patterns could be described. However, this was not compliant with the Journal of Physiology guidelines for statistics policy and therefore our original ms was modified to be compliant with the guidelines, which state:

"P value for statistical tests

For a given conclusion to be assessed, the exact p values must be stated to three significant figures even when 'no statistical significance' is being reported. These should be stated in the main text, figures and their legends and tables. The only exception to this is if p is less than 0.0001, in which case '<' is permitted. Trend statements are not permitted (i.e. 'x increased, but was not significant'). Where there are many comparisons, a table of p values may be appropriate. Asterisks alone should not be used to denote significance within figures.."

Given the many different P values for these time effects, the requirement for exact P values to be stated in the main text, together with the preferred graphical data presentation (commented on positively by the reviewer), we argue that our current approach is required for the ms to be compliant and the detailed approach best explains our complex data set findings. So whilst this referee request seems perfectly reasonable, to make these changes would make the ms non-compliant.

Therefore no changes were made.

Referee#2: Consider moving the section about plasma volume, Hct, BV and Hb to a supplementary section. These measurements are of course important to ascertain that fluid shifts do not affect the observations, but this can be simplified to few sentences and the results be made available further scrutiny in a supplementary file, perhaps in the form of a table.

Response: Thank you for this suggestion. We did previously consider doing exactly this, but note that The Journal of Physiology policy no longer allows publication of such sections or Tables in a supplementary file. Our original submitted ms indeed had such a supplementary file to assist reviewers but this was removed in accordance with Journal of Physiology advice: "*Your paper contains Supporting Information of a type that we no longer publish. Any information essential to an understanding of the paper must be included as part of the main manuscript and figures. The only Supporting Information that we publish are video and audio, 3D structures, program codes and large data files.*" Therefore this request to move the section into supplementary File unfortunately cannot be actioned. For the same reasons as stated in response to the above comments about plasma $[K^+]$, the need to present exact P values to comply with the Journal stats guidelines also means that we can't simply reduce the text to show patterns or make general comments, etc. Finally, as the referee acknowledges the importance of this data, we argue that it isn't appropriate to delete this section from the ms.

Therefore no changes were made.

Referee#2: There is a discussion about the validity of the 3-O-MFPase assay in the methods section. It reads like the authors are a bit uncertain about this method. The finding presented in the results section is a bit paradoxical with respect to the changes during exercise. 3-O-MFPase activity goes down, while ouabain binding goes up. This result is now reported in the results, but is not discussed. I think it would be relevant to discuss this paradoxical finding - or alternatively, if the authors do not believe in the validity of the 3-O-MFPASE measurement, then simply omit these results.

Response: Thank you for this critique. As suggested we have now added a discussion of these activity findings into the discussion section. Given that the depressive effect of exercise on 3-O-MFPase activity has previously been well

described, we restrict this discussion to one additional paragraph, with focus on the apparent paradoxical finding between increased content but reduced activity with exercise. Given previous paragraphs concentrated on the effects of exercise on NKA content, we have not added further commentary on OB in this paragraph. We have also added "and 3-O-MFPase activity" to the subtitle in the Discussion. We have also made small changes to the text in Methods in the 3-O-MFPase activity section in relation to exercise.

Referee#2: Methods p5 still includes a section on the NKA subunit isoforms. I don't think this is necessary, when these measurements have been taken out of the manuscript

Response: We cannot find any remaining section in methods still referring to NKA subunit isoforms. Therefore no changes were made.

Referee#2: Discussion p 26. The section on what level of K⁺ changes can be expected in the interstitium seems a bit speculative. This part could be shortened or omitted.

Response: We agree this is speculative. This was included to demonstrate that the changes in [K⁺] that we observed with exercise could in fact be relevant for muscle fatigue. As such we argue that retention of this point is important. However, consistent with reviewer suggestion, we have shorted this section by ~ 11 lines, by removing unnecessary text and also omitting reference to the Kravtsova paper.

Referee#2: The digoxin occupancy is reported to be around 8%, but when you calculate the difference in digoxin occupancy between PLA and DIG it is actually only a 4% higher occupancy in DIG. Perhaps it could be discussed if this very small difference is a possible explanation for the lack of changes seen in K levels and K fluxes between the two interventions.

Response: The reviewer is correct that the difference in digoxin occupancy between CON and DIG was ~4%. However, the occupancy data are only an indication of the fraction of NKA that has been inhibited by digoxin and don't indicate the functional capacity of the NKA. The most physiologically relevant NKA value reflecting the functional capacity of NKA is OB-Fab, since this represents the number of non-digoxin bound (and therefore functioning) NKA. The OB-Fab was not different between CON and DIG, which we consider to be the most likely reason for the lack of effect of digoxin on [K⁺]. This was already stated in the ms (track change version) in Discussion on P 25 para 2 and also Line 15 in the Conclusion section on Page 27. Therefore no changes were made.

Dear Professor McKenna,

Re: JP-RP-2022-283017R2 "Oral digoxin effects on exercise performance, K⁺ regulation and skeletal muscle Na⁺,K⁺-ATPase in healthy humans" by Simon Sostaric, Aaron Petersen, Craig A Goodman, Xiaofei Gong, Tai Juan Aw, Malcolm J Brown, Andrew Garnham, Collene H Steward, Kate T Murphy, Kate A Carey, James Leppik, Steve F Fraser, David Cameron-Smith, Henry Krum, Rod J Snow, and Michael J McKenna

Thank you for submitting your revised Research Article to The Journal of Physiology. It has been assessed by the original Reviewing Editor and Referees and has been well received. Some final revisions have been requested.

The reports are copied at the end of this email. Please address all of the points and incorporate all requested revisions, or explain in your Response to Referees why a change has not been made.

NEW POLICY: In order to improve the transparency of its peer review process The Journal of Physiology publishes online as supporting information the peer review history of all articles accepted for publication. Readers will have access to decision letters, including all Editors' comments and referee reports, for each version of the manuscript and any author responses to peer review comments. Referees can decide whether or not they wish to be named on the peer review history document.

Authors are asked to use The Journal's premium BioRender (<https://biorender.com/>) account to create/redraw their Abstract Figures. Information on how to access The Journal's premium BioRender account is here: <https://physoc.onlinelibrary.wiley.com/journal/14697793/biorender-access> and authors are expected to use this service. This will enable Authors to download high-resolution versions of their figures. The link provided should only be used for the purposes of this submission. Authors will be charged for figures created on this premium BioRender account if they are not related to this manuscript submission.

I hope you will find the comments helpful and have no difficulty returning your revisions within 2 weeks.

Your revised manuscript should be submitted online using the links in Author Tasks Link Not Available.

Any image files uploaded with the previous version are retained on the system. Please ensure you replace or remove all files that have been revised.

REVISION CHECKLIST:

- Article file, including any tables and figure legends, must be in an editable format (eg Word)
- Abstract figure file (see above)
- Statistical Summary Document
- Upload each figure as a separate high quality file
- Upload a full Response to Referees, including a response to any Senior and Reviewing Editor Comments;
- Upload a copy of the manuscript with the changes highlighted.

- A potential 'Cover Art' file for consideration as the Issue's cover image;
- Appropriate Supporting Information (Video, audio or data set https://jp.msubmit.net/cgi-bin/main.plex?form_type=display_requirements#supp).

To create your 'Response to Referees' copy all the reports, including any comments from the Senior and Reviewing Editors, into a Word, or similar, file and respond to each point in colour or CAPITALS and upload this when you submit your revision.

I look forward to receiving your revised submission.

If you have any queries please reply to this email and staff will be happy to assist.

Yours sincerely,

Scott K. Powers
Senior Editor
The Journal of Physiology
<https://jp.msubmit.net>
<http://jp.physoc.org>
The Physiological Society
Hodgkin Huxley House
30 Farringdon Lane
London, EC1R 3AW
UK
<http://www.physoc.org>
<http://journals.physoc.org>

EDITOR COMMENTS

Reviewing Editor:

The remaining Reviewer still has one minor point to make, which the authors should take into consideration.

Senior Editor:

Comments for Authors to ensure the paper complies with the Statistics Policy:

Given the nature of these data, I do not believe that raw data points are required on the figures because inclusion would likely make the figures more difficult for readers to interpret.

Comments to the Author:

Thank you for the revision of your report. The reviewer and review editor are happy with your revisions. However, the reviewer has asked that you consider one additional (minor) point before acceptance of your paper for publication. We look forward to receiving your revised manuscript.

REFEREE COMMENTS

Referee #2:

The authors have responded adequately to all comments. In some instances the journal policies prevent the changes requested.

In one comment I had written "Methods, p5", but it should have been "introduction, p5". Perhaps the authors will consider whether the section in the introduction about NaK pump subunit isoforms is relevant.

(Comment and response from previous round: Referee#2: Methods p5 still includes a section on the NKA subunit isoforms. I don't think this is necessary, when these measurements have been taken out of the manuscript

Response: We cannot find any remaining section in methods still referring to NKA subunit isoforms. Therefore no changes were made.)

END OF COMMENTS

2nd Confidential Review

22-Jun-2022

JP-RP-2022-283017 "Oral digoxin effects on K⁺ regulation with exercise, fatigue, skeletal muscle Na⁺,K⁺-ATPase and muscle strength in healthy humans" by Sostaric, Petersen et al.

Critique of JP-RP-2022-283017 and Author Responses

Senior Editor: Thank you for the revision of your report. The reviewer and review editor are happy with your revisions. However, the reviewer has asked that you consider one additional (minor) point before acceptance of your paper for publication. We look forward to receiving your revised manuscript.

Response: This point has been addressed below with appropriate changes to the ms.

We hope that this is revised ms is now satisfactory for acceptance of the paper for publication. Many thanks for the careful considerations of the ms by yourself, the reviewing editor and the referees.

Reviewing Editor: The remaining Reviewer still has one minor point to make, which the authors should take into consideration.

Response: This point has been addressed below with appropriate changes to the ms

Referee#2 Critique and Author Response

Referee#2: In one comment I had written "Methods, p5", but it should have been "introduction, p5". Perhaps the authors will consider whether the section in the introduction about NaK pump subunit isoforms is relevant..

Response:

Our apologies for not connecting with the page number that you correctly identified, we focussed on Methods where we had deleted the text previously.

We have now deleted the initial four sentences in para 3 of the Introduction, p5, which specifically referred to NKA isoforms. To ensure that the next sentence still made sense after these deletions and flowed subsequently, we added a few words into the sentence: "against ouabain inhibition in animal..." and also copied the references Nielsen & Clausen, 1996; Clausen, 2013 and Radzyukevich et al, 2013 from the previous, now deleted sentences.

We trust that these changes now meet your requirements and we thank you for your excellent and detailed review of this ms.

Dear Dr McKenna,

Re: JP-RP-2022-283017R3 "Oral digoxin effects on exercise performance, K⁺ regulation and skeletal muscle Na⁺,K⁺-ATPase in healthy humans" by Simon Sostaric, Aaron Petersen, Craig A Goodman, Xiaofei Gong, Tai Juan Aw, Malcolm J Brown, Andrew Garnham, Collene H Steward, Kate T Murphy, Kate A Carey, James Leppik, Steve F Fraser, David Cameron-Smith, Henry Krum, Rod J Snow, and Michael J McKenna

I am pleased to tell you that your paper has been accepted for publication in The Journal of Physiology.

NEW POLICY: In order to improve the transparency of its peer review process The Journal of Physiology publishes online as supporting information the peer review history of all articles accepted for publication. Readers will have access to decision letters, including all Editors' comments and referee reports, for each version of the manuscript and any author responses to peer review comments. Referees can decide whether or not they wish to be named on the peer review history document.

The last Word version of the paper submitted will be used by the Production Editors to prepare your proof. When this is ready you will receive an email containing a link to Wiley's Online Proofing System. The proof should be checked and corrected as quickly as possible.

Authors should note that it is too late at this point to offer corrections prior to proofing. The accepted version will be published online, ahead of the copy edited and typeset version being made available. Major corrections at proof stage, such as changes to figures, will be referred to the Reviewing Editor for approval before they can be incorporated. Only minor changes, such as to style and consistency, should be made a proof stage. Changes that need to be made after proof stage will usually require a formal correction notice.

All queries at proof stage should be sent to TJP@wiley.com

Are you on Twitter? Once your paper is online, why not share your achievement with your followers. Please tag The Journal (@jphysiol) in any tweets and we will share your accepted paper with our 23,000+ followers!

Yours sincerely,

Scott K. Powers
Senior Editor
The Journal of Physiology
<https://jp.msubmit.net>
<http://jp.physoc.org>
The Physiological Society
Hodgkin Huxley House
30 Farringdon Lane
London, EC1R 3AW
UK
<http://www.physoc.org>
<http://journals.physoc.org>

P.S. - You can help your research get the attention it deserves! Check out Wiley's free Promotion Guide for best-practice recommendations for promoting your work at www.wileyauthors.com/eeo/guide. And learn more about Wiley Editing Services which offers professional video, design, and writing services to create shareable video abstracts, infographics, conference posters, lay summaries, and research news stories for your research at www.wileyauthors.com/eeo/promotion.

*** IMPORTANT NOTICE ABOUT OPEN ACCESS ***

To assist authors whose funding agencies mandate public access to published research findings sooner than 12 months after publication The Journal of Physiology allows authors to pay an open access (OA) fee to have their papers made freely available immediately on publication.

You will receive an email from Wiley with details on how to register or log-in to Wiley Authors Services where you will be able to place an OnlineOpen order.

You can check if your funder or institution has a Wiley Open Access Account here <https://authorservices.wiley.com/author-resources/Journal-Authors/licensing-and-open-access/open-access/author-compliance-tool.html>

Your article will be made Open Access upon publication, or as soon as payment is received.

If you wish to put your paper on an OA website such as PMC or UKPMC or your institutional repository within 12 months of publication you must pay the open access fee, which covers the cost of publication.

OnlineOpen articles are deposited in PubMed Central (PMC) and PMC mirror sites. Authors of OnlineOpen articles are permitted to post the final, published PDF of their article on a website, institutional repository, or other free public server, immediately on publication.

Note to NIH-funded authors: The Journal of Physiology is published on PMC 12 months after publication, NIH-funded authors DO NOT NEED to pay to publish and DO NOT NEED to post their accepted papers on PMC.

EDITOR COMMENTS

Reviewing Editor:

The authors responded satisfactorily to the remaining comment.

Senior Editor:

Thank you for submitting your work to the Journal of Physiology. Congratulations on the completion of an outstanding study.